# Yose-Ue: A Treap-Based Ensemble Framework for Resource-Efficient Unsupervised Anomaly Detection

## Abstract

Anomaly detection seeks to identify observations that deviate significantly from an underlying data distribution. While deep learning and ensemble-based approaches have achieved strong empirical performance, their computational and memory requirements limit their applicability in resource-constrained edge environments. Furthermore, many approaches to improving efficiency rely on supervised models, which require labeled anomalies that are often scarce in practice. We propose Yose-Ue, a resource-efficient, fully unsupervised anomaly detection framework based on treap-structured ensemble learning. Yose-Ue co-designs a compact data representation with computationally efficient split-selection mechanisms. Specifically, we construct a randomized treap (a hybrid tree–heap data structure) in which nodes are defined by discretized split points, and priorities are assigned via a mass-driven criterion that favors informative partitions. This design yields balanced hierarchical partitions while maintaining low memory overhead. The resulting ensemble estimator improves structural diversity and statistical robustness without incurring the computational cost typical of deep architectures. We provide a comparative evaluation against established unsupervised ensemble baselines (Isolation Forest, DiForest, and EXTiForest) and resource-efficient state-of-the-art methods, including AutoEncoder, Graph Attention Network AutoEncoder, Histogram-Based Outlier Score, and Local Outlier Factor. Experiments conducted on 14 benchmark datasets—including synthetic datasets, experimental mobile-sensor data, and datasets from the ODDS repository—demonstrate that Yose-Ue achieves competitive or superior detection performance while substantially reducing computational complexity. The proposed method attains over $126\times$ reduction in training time and $7\times$ reduction in inference latency relative to representative baselines. These results indicate that treap-based ensemble learning provides a principled and scalable approach to unsupervised anomaly detection in edge-constrained environments.

## 1 Introduction & Motivations

Anomaly detection is a fundamental task that involves identifying and isolating abnormal data points (Steinwart et al., 2005; Chandola et al., 2009; Nassif et al., 2021; Bouman et al., 2024). Traditional statistical approaches to anomaly detection are grounded in hypothesis testing and parametric modeling (Neyman & Pearson, 1933; Grubbs, 1969; Basseville & Nikiforov, 1993; Urvoy & Autrusseau, 2014; Kamenik & Szewc, 2023). These methods provide interpretability and theoretical guarantees but rely on strong distributional assumptions, such as the data following a Gaussian distribution with mean $\mu$ and variance $\sigma^2$. Recent advances in machine learning (ML) relax these assumptions by learning representations of normal behavior directly from data in a flexible, model-agnostic manner. Such approaches employ sophisticated algorithms to uncover complex statistical structures inherent in the data (Liu et al., 2008; Wankhede, 2019; Xu et al., 2023; Hariri et al., 2021; Ortega et al., 2025). By modeling these underlying patterns, ML-based methods can detect deviations that indicate potential anomalies. Deep learning, particularly Deep Neural Networks (DNNs), further enhances this flexibility by capturing highly nonlinear and hierarchical structures in complex, high-dimensional datasets (Tran et al., 2023; Bouniot et al., 2025).

As a result, DNN- and ML-based anomaly detection methods have been widely adopted in compute-intensive domains, including intrusion detection, network monitoring, cloud infrastructure, cyber-physical systems, and fault diagnosis (Nassif et al., 2021; Pang et al., 2021; Ramson et al., 2020; Han et al., 2016; Yao et al., 2017; Guo et al., 2016; Kulkarni & Sinha, 2012; Kumar et al., 2017; Daghero et al., 2024). However, many real-world applications—such as sensor fault detection (Alwaisi et al., 2024), security monitoring (Kalør et al., 2021), and predictive maintenance (Mennilli et al., 2025)—require systems to operate reliably under strict computational and memory constraints. Although DNNs provide powerful modeling capabilities, their inference phase demands substantial computational resources, necessitating careful optimization for efficient deployment (Wu et al., 2023; Rhu et al., 2016). While server-class platforms typically offer sufficient resources (Wu et al., 2023; Rhu et al., 2016; Nvidia, 2025), edge devices often lack the capacity to support such workloads. Consequently, deploying DNN-based anomaly detection on resource-constrained platforms is fundamentally challenging (Wu et al., 2023; Rhu et al., 2016; Nvidia, 2021; Espressif, 2025). For example, a typical microcontroller may provide only 120 KB of memory, rendering modern DNN inference impractical without complex and difficult-to-integrate compression or optimization strategies (Wu et al., 2023; Rhu et al., 2016; Espressif, 2025; Saha et al., 2022). Despite these limitations, resource-constrained devices remain attractive for inference applications due to their low cost and widespread availability (Boi et al., 2025; Saha et al., 2022; Deutel et al., 2025; Coelho et al., 2021; Alwaisi et al., 2024; Kalør et al., 2021; Mennilli et al., 2025). Existing approaches are either inherently resource-intensive (e.g., DNNs) or lack a principled mechanism for balancing detection efficacy and computational efficiency. This gap motivates the co-design of memory-efficient data representations and computationally efficient algorithms for effective anomaly detection in constrained environments.

This paper addresses this need. We propose Yose-Ue, a method named after the Japanese forest-bundling technique, which constructs a carefully designed ensemble of estimators for anomaly detection. Each estimator in the Yose-Ue ensemble is a treap (Seidel & Aragon, 1996), a hybrid data structure that combines properties of a random binary search tree and a max-heap. The core contribution of this work is a novel treap-based estimator that leverages a max-inverse-mass principle in two ways: (1) to guide treap growth through split-point selection, and (2) to enable a mass-aware anomaly scoring function.

During training, treap split points are selected by discretizing feature domains. This discretization effectively constructs histograms over relevant features, allowing the estimation of the "mass" of nominal data regions. Regions with higher mass correspond to higher data density and are therefore considered more normal than low-mass regions. Each estimator enforces a max-inverse-mass priority during treap construction, ensuring that low-mass (i.e., high inverse-mass) regions are partitioned earlier in the representational structure. Each treap is regularized through the training data, discretization scheme, and an estimation heuristic specified by input parameters. Through this joint data structure–algorithm co-design, Yose-Ue achieves efficient, ensemble-based unsupervised anomaly detection. To approximate a multivariate mass function, each treap introduces randomness by selecting features at each split. The resulting multivariate mass estimate serves as the anomaly scoring function. During inference, a test sample traverses each treap from the root to a leaf node. The split points along the traversal path are accumulated to infer the mass of the region from which the test point originates, yielding a mass-aware anomaly score. This inference procedure is both interpretable and grounded in the underlying data distribution. Importantly, interpretability provides actionable insight into model decisions, which is essential in mission-critical anomaly detection scenarios. In contrast, DNN-based anomaly detection methods are typically derived from black-box representations and therefore lack inherent interpretability (Chen et al., 2018; Salehi & Davulcu, 2020).

In conjunction, the proposed ensemble technique advances state-of-the-art ensemble methods (SOTA) that take advantage of tree-based structures to represent nominal data distributions, such as iForest (Liu et al., 2008). Prior SOTA approaches use tree path depth as a proxy for distributional density, whereas the proposed method introduces a mass-estimated path-traversal score. Figure 1 illustrates the comparison between the SOTA path-depth estimate and the proposed mass-based estimate for a univariate Gaussian distribution. As shown, the SOTA method produces noisy estimates of path-depth in the nominal region of the toy Gaussian distribution (near $\mu$). In contrast, the proposed mass-based estimates yield a more accurate characterization

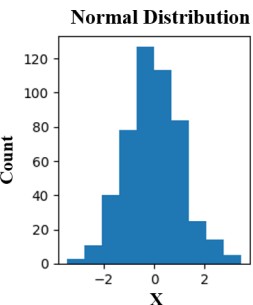 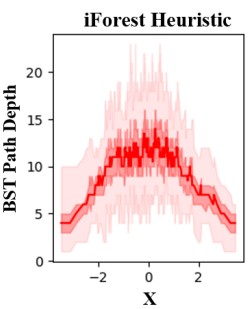 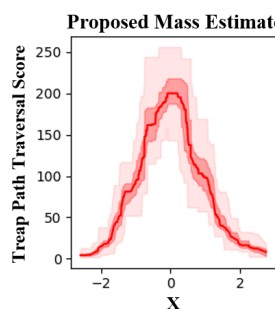

Figure 1: Empirical comparison between iForest path depth heuristic vs proposed mass estimate for anomaly scoring across a normal distribution. Only 25 estimators are permitted per ensemble. Each estimator is constructed from 256 sub-sampled data points.

of the nominal density. We demonstrate that this improvement in nominal estimation systematically enhances anomaly detection performance, particularly for resource-constrained small ensembles[1].

The major contributions of this paper are as follows:

1. We introduce a novel treap ensemble method that serves as a multivariate mass-estimation proxy for anomaly scoring.

2. We demonstrate up to a $10\times$ improvement in Kolmogorov–Smirnov goodness-of-fit test performance over previous SOTA ensemble-based distribution estimates, using a Gaussian kernel density estimator as a reference.

3. Across a diverse evaluation suite—including three baseline ensemble methods, four SOTA unsupervised anomaly detection methods, and multiple datasets (three synthetic datasets, three experimental iPhone datasets, six standard anomaly detection benchmarks, and two benchmarks with dimensionality reduction)—Yose-Ue either improves or maintains competitive detection performance, while providing enhanced memory efficiency and significant training-time and inference-speed improvements relative to SOTA methods.

The remainder of this paper is organized as follows. Section 2 reviews related work. Section 3 formally defines the problem addressed in this study. Section 4 details the proposed method. Section 5 offers a complexity analysis of the treap-learner framework. Section 6 describes the datasets and experimental setup. Section 7 presents a design space exploration, Section 8 provides a quantitative comparison for distributional representation, Section 9 details baseline comparison of three binary search tree ensemble methods and the proposed treap learner framework, Section 10 compares the contributions of the proposed training and inference techniques to SOTA forest-based ensembles, and Section 11 expands our comparison with a comprehensive evaluation of Yose-Ue against four SOTA methods with fourteen datasets. Finally, Section 12 concludes the paper.

## 2 Related Prior Work

In practice, anomalous events occur infrequently and are heavily outnumbered by nominal instances (Studiawan & Sohel, 2020), making the collection of labeled anomaly data costly and impractical. Although semi-supervised and supervised techniques designed for imbalanced data have been proposed to address these challenges (Han, 2020; Yoon et al., 2023; Hien et al., 2024; Qiao et al., 2024; Finke et al., 2024; Ai et al., 2025; Durani et al., 2025), this work focuses on the unsupervised setting, which is widely adopted in anomaly detection. In this section, we review relevant prior work along two primary directions. First, we examine existing methods for unsupervised anomaly detection. Second, we discuss prior efforts aimed at reducing the memory footprint of machine-learning models to enable efficient inference on resource-constrained platforms.

---

[1]See our code and scripts that helped generate all experiments and figures: `https://anonymous.4open.science/r/Yose-Ue-11DB/README.md`

## 2.1 Unsupervised Anomaly Detection

### 2.1.1 Ensemble Methods

Forest-based ensemble methods are used for unsupervised anomaly detection due to their ability to balance accuracy with quasi-linear compute and memory complexity (LinkedIn, 2025; MathWorks, 2025; **?**; Melquiades & de Lima Neto, 2022; Hariri et al., 2021; Ma et al., 2020; Ortega et al., 2025; Xu et al., 2023). A notable method in this domain is the Isolation Forest (iForest), which isolates anomalies by recursively partitioning data through randomly grown binary search trees (BSTs). Several adaptations of IF have been proposed to improve performance or extend its capabilities. Hybrid Isolation Forest is a semi-supervised learning method that incorporates synthetic labels to guide anomaly scoring (Melquiades & de Lima Neto, 2022). However, its reliance on labeled anomalies restricts its practical usability.

Extended Isolation Forest (EXT-iForest) improves detection accuracy by introducing hyperplane-based splits and oblique cuts; however, these enhancements incur additional computational and memory overhead, limiting suitability for edge deployment (Hariri et al., 2021). Similarly, Isolation Mondrian Forest integrates iForest with Mondrian Forests to enable online learning capabilities (Ma et al., 2020).

Discretized Isolation Forest (DiForest) proposes a compute- and memory-efficient extension of iForest tailored for unsupervised anomaly detection on edge devices (Ortega et al., 2025). By discretizing split-point candidates, DiForest preserves several core strengths of iForest while reducing memory requirements. Deep Isolation Forest (DEEP-iForest) further extends the framework by incorporating a neural network for spatial reconstruction alongside a back-end iForest (Xu et al., 2023). This hybrid design generalizes both iForest and EXT-iForest by enabling nonlinear data partitions, resulting in improved anomaly detection performance.

A key methodological difference between these past iForest variants and Yose-Ue lies in their anomaly-scoring mechanisms. All of the aforementioned iForest variants use a "path depth" heuristic in their anomaly-detection scoring mechanism: the deeper the path, the more nominal a point is. We show later that this widely used heuristic yields poor nominal estimates relative to the ground-truth nominal data. In our proposed approach, we show that deriving treap estimators with informed split points and scoring based on path traversals yields improved nominal representation.

### 2.1.2 DNN Methods

Although a wide range of DNN-based methods have been applied to anomaly detection (Chen et al., 2018; Goodfellow et al., 2014; Salehi & Davulcu, 2020; Tsai et al., 2025), we focus on four primary architectures commonly used in unsupervised settings. The first is a foundational DNN architecture: the autoencoder (Chen et al., 2018). The second is the generative adversarial network (GAN) (Goodfellow et al., 2014). The third and fourth architectures are more recent approaches, namely Graph Attention Networks (GATs) (Salehi & Davulcu, 2020) and Large Language Models (LLMs) (Tsai et al., 2025).

A classical example of an unsupervised DNN for anomaly detection is the autoencoder (Chen et al., 2018). An autoencoder comprises two components: an encoder, which compresses the input into a lower-dimensional latent representation, and a decoder, which reconstructs the original input from this latent space. This architecture forms the foundation of numerous anomaly detection approaches (Zhou & Paffenroth, 2017; Yao et al., 2019). Building upon the autoencoder paradigm, the generative adversarial network (GAN) introduces a fundamentally different framework. A GAN consists of two neural networks—a generator and a discriminator—trained jointly in an adversarial setting to generate and distinguish data representations (Goodfellow et al., 2014; Schlegl et al., 2017). During inference, the learned model is used to assess whether a test sample aligns with the distribution of the training data. Although GAN-based approaches have significantly advanced DNN capabilities for anomaly detection (Lüer & Bohm, 2024; Wang et al., 2018; Dehghanian et al., 2024), their substantial memory and computational requirements often render them impractical for deployment on resource-constrained platforms (Liu et al., 2023; Jayakodi et al., 2020; Kumar et al., 2023).

Graph attention networks (GATs) have emerged to address the challenges of learning from graph-structured data (Salehi & Davulcu, 2020). GATs leverage attention mechanisms to weigh the importance of neighboring nodes when aggregating information, enabling flexible and effective representation learning (Salehi

& Davulcu, 2020). This approach enhances DNN's ability to capture complex relationships in graph data. These learning mechanisms have become increasingly popular for unsupervised anomaly detection (Zhao et al., 2020; Latif-Martínez et al., 2025; Liu et al., 2024). Recently, an LLM method was used to perform anomaly detection on tabular datasets (Tsai et al., 2025). It is important to note that deploying this method typically requires server-grade computational resources (Tsai et al., 2025).

## 2.2 Reducing Memory Footprint for Inference

Emerging edge applications are constrained by limited compute and memory resources on resource-constrained devices (Saha et al., 2022). To address this challenge, recent research has focused on adapting ML method inference for resource-constrained environments (Wu et al., 2023; Han et al., 2016; Yao et al., 2017; Guo et al., 2016; Kulkarni & Sinha, 2012; Kumar et al., 2017; Daghero et al., 2024; Boi et al., 2025; Saha et al., 2022). To enable DNN inference in such environments, prior work focuses on reducing the memory footprint through techniques such as compression, quantization, and pruning (Wu et al., 2023; Han et al., 2016; Yao et al., 2017; Guo et al., 2016; Saha et al., 2022). In addition, these techniques are typically applied post hoc to mitigate inefficiencies discovered after DNN training.

Parallel efforts have focused on adapting tree-based machine learning algorithms—such as Random Forests (RF) and Gradient-Boosted Trees (GBT)—for edge deployment (Kulkarni & Sinha, 2012; Kumar et al., 2017). These approaches reduce inference costs through techniques such as tree pruning and sparsity enforcement (Kulkarni & Sinha, 2012; Kumar et al., 2017). While effective, they assume access to labeled data and operate within a supervised learning framework, limiting their applicability to anomaly detection scenarios where labeled anomalies are scarce.

In contrast, the proposed approach is inherently efficient for small ensembles. Specifically, Yose-Ue reduces the training search space of each ensemble estimator through a tailored discretization scheme and estimator-selection heuristic. By embedding computational and memory efficiency directly into the method's design, the proposed framework minimizes the need for complex post hoc optimization and facilitates deployment in resource-constrained inference environments. In this context, the proposed Yose-Ue provides a practical design/evaluation framework in which memory efficiency, interpretability, and competitive detection performance are jointly important.

## 2.3 Efficient Learning for Edge Applications

Although this work focuses on ultra-efficient inference, the same efficiency principles can also support edge-specific training and fine-tuning. In many deployed settings, the bottleneck is not only inference latency, but also the ability to adapt models under strict energy, memory, bandwidth, and compute limits. Remote sensing provides a clear example: tasks such as wildfire detection, disaster response, and hyperspectral anomaly detection require timely decisions when transmitting full-resolution data to the cloud may be costly or infeasible. Onboard AI systems such as ESA's Φsat-2 demonstrate the value of in-orbit processing for applications such as wildfire detection (ESA, 2025a;b), while onboard preprocessing and data-prioritization methods reduce downlink cost by filtering or selecting useful imagery before transmission (2024Chatar, Keenan and Kitamura, Kentaro and Cho, Mengu20242018; Qi, Baogui and Shi, Hao and Zhuang, Yin and Chen, He and Chen, Liang2018). However, remote sensing data can vary across sensors, geographies, seasons, atmospheric conditions, illumination, and event types. This creates a need for lightweight local adaptation, where efficient training or fine-tuning can update deployed models without requiring full cloud-scale retraining (2026Marti Escofet, Francesc and Blumenstiel, Benedikt and Scheibenreif, Linus and Fraccaro, Paolo and Schindler, Konrad2026).

# 3 Anomaly Detection Preliminaries

Let $\mathcal{X} \subset \mathbb{R}^d$ denote an unlabeled dataset containing nominal (normal) data points:

$$\mathcal{X} = \{x_i\}_{i=1}^{n}, x_i \in \mathbb{R}^d. \tag{1}$$

We assume that an underlying distribution generates these points. Let $p_{\mathcal{X}}$ be this distribution on $\mathbb{R}^d$, and let $x_1, ..., x_n$ be drawn i.i.d from this distribution. Note that $p_{\mathcal{X}}$ is unknown. The goal of unsupervised anomaly detection is to define a score function $s : \mathbb{R}^d \rightarrow [0, 1]$ that measures how anomalous a test point $x \in \mathbb{R}^d$ is, relative to the nominal distribution $p_{\mathcal{X}}$. For example, consider two test points $x_{\text{nominal}} \sim x_i$ and $x_{\text{anomaly}} \nsim x_i$. The ideal anomaly scoring function $s$ would provide the following classification: $s(x_{\text{nominal}}) = 0$ and $s(x_{\text{anomaly}}) = 1$. Specifically, higher values of $s(x)$ indicate points that lie in anomalous low-density regions of $p_{\text{nominal}}$, and vice versa.

The central challenge in unsupervised anomaly detection is to accurately and efficiently estimate the underlying data distribution, $p_{\mathcal{X}}(x)$. However, for anomaly detection tasks, precise estimation of the full data distribution is often unnecessary. In practice, detection primarily depends on identifying deviations from nominal structure—such as changes in relative ordering, local mass concentration, or partition occupancy.

Full density estimation is inherently difficult and may introduce unnecessary modeling complexity without corresponding gains in detection performance. These considerations motivate approaches that operate over adaptive, data-dependent partitions of the feature space, where nominal behavior is characterized by relative mass and ordering rather than explicit probability density estimates. Accordingly, the proposed method is designed to address the following problem.

**Problem Definition:** Let $\mathbb{R}^d$ denote the feature space and let $p_{\mathcal{X}}$ denote the unknown nominal data distribution generating our dataset $\mathcal{X} \subset \mathbb{R}^d$. Given a set of nominal $n$ samples, $\mathcal{X} = \{x_i\}_{i=1}^n$, the goal of unsupervised anomaly detection is to learn a scoring function $s : \mathbb{R}^d \rightarrow [0, 1]$ such that instances lying on low-probability (low-mass) regions of $p_{\mathcal{X}}$ receive higher anomaly scores than instances in high-mass regions. Formally, the scoring function $s$ should preserve the inverse ordering induced by the nominal distribution, i.e., if $p_{\mathcal{X}}(x_{nominal}) \geq p_{\mathcal{X}}(x_{\text{anomaly}})$ then $s(x_{\text{nominal}}) \leq s(x_{\text{anomaly}})$.

## 4 Yose-Ue

We begin by formally introducing the proposed method and presenting the key insights underlying this approach. We then define its scope and describe the associated discretization, training, and inference procedures. Next, we derive the computational complexity of the approach. Finally, we provide insights into the roles of global and local estimation within the proposed treap ensemble framework.

### 4.1 Preliminaries

Given a dataset $\mathcal{X} = \{x_i\}_{i=1}^n \subset \mathbb{R}^d$, we proceed as follows:

**Discretization of Input Features:** For each dimension $f \in \{1, 2, \ldots, d\}$, we partition the feature value range into $\alpha$ uniform bins, according to a selected discretization rule. Further information on the discretization rules used in this work is discussed in Section 4.3. Each bin is associated with an empirical mass estimate and a split point candidate, i.e., the number of data points that fall into that bin and the center of that bin. A bin's split point candidate is the center of the bin, referred to as $c_k^f$, and is defined in Section 4.3. The bin's mass estimate, $m_k^f$, is the bin count and is defined as follows.

$$m_k^f = \sum_{i=1}^n \mathbf{1}\left\{x_i[f] \in \text{bin } k\right\} \tag{2}$$

The bin mass and bin center points per feature are used for the proposed method. These are represented by the arrays $B_{\text{array·f}}$ and $B_{\text{count·f}}$, defined below.

$$B_{\text{array·f}} = \{c_1^f, \ldots, c_k^f, \ldots, c_\alpha^f\} \tag{3}$$

$$B_{\text{count·f}} = \{m_1^f, \ldots, m_k^f, \ldots, m_\alpha^f\} \tag{4}$$

Note that $B_{\text{array·f}}$ and $B_{\text{count·f}}$ are constructed per feature. We use $B_{\text{array}}$ and $B_{\text{count}}$ to encapsulate all split point candidates and mass estimates for features $f \in \{1, 2, \ldots, d\}$, such that $B_{\text{array·f}} \in B_{\text{array}}$ and $B_{\text{count·f}} \in B_{\text{count}}$.

**Formalization:**  The proposed method, Yose-Ue, is a treap ensemble. A treap ensemble is a collection of hybrid BST-plus-heap estimators (treaps), each trained independently on subsampled data from the original training set. Yose-Ue generates $N_{\text{treaps}} > 0$ treaps to partition our considered dataset $\mathcal{X}$. To create a treap, a subsample of points from $\mathcal{X}$ is randomly selected to construct a single treap. The number of subsampled points is $\psi > 1$. This is referred to as our subsampling rate. The subsampled set of training data, referred to as $X_\psi$, is a subset of the original training dataset, i.e., $X_\psi \subset \mathcal{X}$. The proposed method supports constructing $B_{\text{count}}$ and $B_{\text{array}}$ from either $X_\psi$ or $\mathcal{X}$. We refer to this as either local estimation or global estimation. This estimation is set as a Boolean input hyperparameter. We provide further information in Section **??**.

The construction of a treap begins by splitting our subsampled training data $X_\psi$. A random feature $f$ from $X_\psi$ is selected. Note that for brevity, we refer to an arbitrarily selected feature as $x_f$. The total number of possible features or dimensions within $\mathcal{X}$ is referred to as $d$. The split point candidates $B_{\text{array}}$, and the mass estimates $B_{\text{count}}$ are filtered based on $x_f$. Note that subsampling induces a retained index set from the candidate split point array $B_{\text{array}}[f]$, i.e., $I_f \subseteq \{1, \ldots, \psi\}$. As both arrays are aligned elementwise, we apply the same restrictions to both arrays, ensuring that each retained candidate preserves its associated mass.

$$B_{\text{array·filter}} := B_{\text{array}}[f][I_f] \tag{5}$$
$$B_{\text{count·filter}} := B_{\text{count}}[f][I_f] \tag{6}$$

We select the index $i$ that maximizes the inverse mass priority $(1/m_k^f)$ from $B_{\text{count·filter}}$. We use index $i$ to identify the split point priority value $(B_{\text{count·filter}}[i])$ and split point candidate value $B_{\text{array·filter}}[i]$, referred to as $s_{\text{val}}$. If the split point $s_{\text{val}}$ does not partition $X_\psi$, a leaf node is returned. If a split is achievable, the data is partitioned by $s_{\text{val}}$ in $x_f$. Each partition is used to continue the treap splitting process recursively. This process continues until all points within $X_\psi$ are isolated or until no split point candidates remain. Once complete, the Yose-Ue treap is returned. An example of this procedure is shown in Figure 2 in Section 4.4. Note that all treaps split nodes retain the split-point candidate $s_{\text{val}}$, the selected feature $f$, and the inverse-mass priority value $1/m_k^f$. This will repeat $N_{\text{treaps}}$ times until all the specified number of treaps are constructed, i.e., to construct the Yose-Ue treap ensemble. A procedure to construct the treap ensemble is provided in Figure 3 in Section 4.4.

When performing inference, the goal is to score a test point $x_{\text{test}}$ if it is within the nominal $\chi$. Before inference, a $p_{\text{prior}}$ variable is constructed and set to 0. During inference, $x_{\text{test}}$ is passed to the root node of a treap, i.e., the first split node. The $p_{\text{prior}}$ variable is incremented by the split node's inverse priority, i.e., $m_k^f$. In addition, the split node's selected feature and its associated split point value are used to determine whether $x_{\text{test}}[f] < s_{\text{val}}$ is true. If the inequality with the test point is true, then the test point is passed to the left child (and to the right child if false). This inference procedure, which we refer to as path-traversal, continues until the test point reaches a leaf node. Note that, as the test point traverses the treap, at each split node the $p_{\text{prior}}$ variable is incremented by the relative inverse-priority (mass). An example of the path traversal inference procedure is shown in Figure 4 in Section 4.4. By averaging the mass estimates across all estimators in the treap ensemble, we obtain the multivariate mass estimate for $\mathcal{X}$ from the test point $x_{\text{test}}$. The mass estimates drive the proposed anomaly-scoring statistic. The mass estimates are represented as a set returned by the function $s_{\text{paths}}(x_{\text{test}})$. We denote the mass estimate from each Treap $i$ by $s_i(\cdot)$.

$$s_{\text{paths}}(x_{\text{test}}) = \{s_i(x_{\text{test}}), i \in [1, \ldots, N_{\text{treaps}}])\} \tag{7}$$

The expected mass estimate from $s_{\text{paths}}$ is subsequently normalized based on the type of discretized estimation used, i.e., $\psi$ for local and $|X|$ for global estimation. This normalized expected mass estimate, denoted $p_{\text{prob}}$, is used for the proposed anomaly-scoring statistic $s(\cdot)$, which is defined as follows.

Table 1: Discretization functions used in this work.

| Discretization Function (De La Rubia, 2024) | $\alpha$ given $x_f$ | Abbrevation |
|---|---|---|
| Sturge's Rule | $\lceil \log_2(|x_f|) + 1 \rceil$ | Sturge |
| Freedman Diaconis' Rule | $\left\lceil \frac{|\max x_f - \min x_f|}{2 \cdot IQR(x_f) \cdot |x_f|^{-1/3}} \right\rceil$ | FD |

$$p_{\text{prob}}(x_{\text{test}}) = \begin{cases} \mathbb{E}[s_{\text{paths}}(x_{\text{test}})]/|X|, & \text{if estimation is global} \\ \mathbb{E}[s_{\text{paths}}(x_{\text{test}})]/\psi, & \text{if estimation is local} \end{cases} \tag{8}$$

$$s(x_{\text{test}}) = 2 - 2^{p_{\text{prob}}(x_{\text{test}})} \tag{9}$$

A pseudocode example of this inference procedure is shown in Figure 6. We clarify and provide further insights on the proposed training and inference procedures in Section 4.4.

## 4.2 Improving SOTA & Our Key Insights

Prior ensemble anomaly detection methods, such as iForest and its variants, utilize binary trees constructed from randomly sampled hyperplane cuts to model nominal data distributions (LinkedIn, 2025; MathWorks, 2025; Liu et al., 2008; 2012; Melquiades & de Lima Neto, 2022; Hariri et al., 2021; Ma et al., 2020; Ortega et al., 2025; Xu et al., 2023). Randomness aids in ensemble diversity. However, uninformed split selection introduces substantial variance in anomaly scores, particularly in dense data regions, where more consistent structure should be preserved. Subsequently, the conventional reliance on tree path depth as an anomaly-scoring heuristic can be misaligned with the true data density. Our proposed method addresses this limitation by incorporating principled, mass-aware split selection via treaps, i.e., binary search trees endowed with max-heap properties. We bias split decisions toward low-density (high inverse-mass) regions, improving nominal structural representations across the ensemble. In place of path depth, we introduce a treap path-traversal score that better captures the rarity of a data point, leading to smoother, more reliable anomaly scoring, particularly in high-density regions. By unifying discretized input domains with data-structure-informed estimation, Yose-Ue offers a scalable, memory-efficient solution for unsupervised anomaly detection.

A key property of the treap Ensemble framework is that the inverse-mass max split rule implicitly preserves reverse density ordering. By focusing on inverse-mass max split-point candidates, the treap estimators preferentially partition the feature space at locations of maximal mass contrast (or isolation gain). Concretely, the split point is chosen to maximize a criterion proportional to the separation between empirical cumulative distributions (or equivalently, to maximize the imbalance in point counts across the partition). This causes regions of high sample density to be recursively subdivided more finely, since dense regions admit many candidate splits with high gain, whereas sparse regions terminate quickly due to low achievable gain. As a result, the proposed path traversal score is implicitly related to both traversal depth and local mass structure. Points in denser regions tend to remain in the treap longer and accumulate higher traversal scores, whereas low-density anomalies are more likely to be separated earlier and receive lower scores. Thus, lower traversal scores generally correspond to higher anomaly scores and lower estimated local mass, without requiring an explicit density estimate. Since the max-priority rule is determined by the empirical mass distribution within each node, it reduces the variability introduced by random splits and promotes a hierarchy that better reflects the observed data geometry. In this sense, path depth is not used as a standalone isolation metric; rather, it is implicitly reflected through the proposed traversal score, whose priority-based construction smooths the ordering of the empirical mass estimate.

Finally, we note that the treap ensemble's ability to resolve complex distributional structures—particularly sharp or multimodal densities—is governed by the discretization strategy used to generate candidate split points. While the inverse-mass max split rule ensures a consistent reverse-density ordering within each node, the resolution of this ordering depends on the discretized feature space. Data-driven discretization schemes (e.g., Sturges or Freedman–Diaconis) control this fidelity: finer partitions enable better separation of narrow, high-density modes. Consequently, regions that appear structurally shallow under coarse discretization do not indicate a limitation of density modeling but reflect the chosen partition resolution.

```
 1: function GROWTHETREAP(B_array, B_count, D_max, D_curr, X)
 2:     if D_curr ≥ D_max or |X| ≤ 2 then
 3:         return Leaf Node
 4:     end if
 5:     Randomly select feature from X
 6:     Filter B_array and B_count based on random feature's range
 7:     Select the bin index (s_idx) that maximizes the inverse mass from the filtered B_count
 8:     Assign the split point (s_val) from the corresponding bin from the filtered B_array
 9:     if s_val does not actually splits X then
10:         return Leaf Node
11:     end if
12:     Set p as the inverse mass priority (1/(B_count·f[s_idx] + ζ))
13:     Partition your data X from the random feature and s_val
14:     Create two children treaps recursively on the partitioned data
15:     return Split Node with selected feature, s_val, p, and the children treaps
16: end function
```

Figure 2: Training algorithm used to row a single treap

## 4.3 Discretization

Yose-Ue provides an efficient data structure through discretization. It computes a discretized set of candidate split points for each feature. The discretization function returns the number of bins, referred to as $\alpha$, to be used for each feature in $X$. Note that the discretization functions considered in this work are Sturge's rule and the Freedman Diaconis' rule (De La Rubia, 2024). Their definition and used abbreviations in this work are noted in Table 1. For a given $x_f$ and discretization rule giving $\alpha$, we derive the bin width, $\delta$. The bin width is computed from dividing the total range of $x_f$ by $\alpha$, i.e., $|\max x_f - \min x_f|/\alpha$. The bin width is defined as follows.

$$\delta = \frac{|\max x_f - \min x_f|}{\alpha} \tag{10}$$

The bin width, $\delta$, is then used to construct an array of bin edges from $x_f$. The bin edges are used to construct a histogram of the feature $x_f$. The bin edge array is denoted by $b$ and is defined as follows.

$$b_f = \{\min x_f + (i \cdot \delta)\}, i \in \{0, \ldots, \alpha\} \tag{11}$$

The centers of these bin edges are the considered split-point candidates. The array of bin edges is used to construct the split-point candidates for a given feature $x_f$. The split point candidate $c_k^f$ is described in Equation 12. Note that subscript $k$ is used to indicate bin index with the following constraints $k \in \{1, \ldots, \alpha\}$.

$$c_k^f = \frac{b_f[k-1] + b_f[k]}{2} \tag{12}$$

The array of split point candidates is $B_{\text{array·f}} = \{c_1^f, \ldots, c_\alpha^f\}$, such that $B_{\text{array·f}} \in B_{\text{array}}$ for $f \in \{1, \ldots, d\}$. As constructed previously in Section 4.1, the associated bin counts $B_{\text{count}}$ follow the split point candidates structure. These constructed split values ($B_{\text{array}}$) and count masses ($B_{\text{count}}$) are used to create our proposed treap structure. The arrays constrain split evaluations to occur only at the max-inverse mass-discretized boundaries, thereby significantly reducing computational and memory overhead.

## 4.4 Training & Inference

**Training:** The training process for a given treap is described in Figure 2 and Figure 3. To construct a treap, a random feature is selected. The $B_{\text{array}}$ and $B_{\text{count}}$ arrays are filtered to reduce the split point search space. A $\arg\max$ strategy is employed to select the split point that maximizes the inverse-mass and subsequently preserves the heap property. This ensures that each parent node has a greater priority (inverse mass) than its children. The inverse-mass priority split secures that lower-mass data partitions are split earlier rather than deeper in the treap. After verifying correctness, the construction process recursively grows the treap until splitting is no longer possible.

```
 1: function GROWTHEENSEMBLE(Disc. Func., Est. Rule, ψ, X, and N_treaps)
 2:     Initiate Yose-Ue as an empty treap ensemble set
 3:     D_max ⇐ ψ − 1
 4:     if Est. Rule is Global then
 5:         Set B_array & B_count per the Disc. Function given X
 6:     end if
 7:     for i = 1 to N_treaps do
 8:         X_ψ ⇐ Sample ψ points from X
 9:         if Est. Rule is Local then
10:             Set B_array & B_count per Disc. Function from X_ψ
11:         end if
12:         Construct treap and add to Yose-Ue
13:     end for
14:     return Yose-Ue
15: end function
```

Figure 3: Training Algorithm for the Yose-Ue Ensemble

```
 1: function COLLECTPATHPRIORITY(Curr. Node, x_test, p_prior)
 2:     Increment p_prior by inverse priority of the Curr. Node
 3:     if Curr. Node is a leaf then
 4:         return p_prior
 5:     end if
 6:     if x_test feature value < Curr. Node split feature value then
 7:         Set next node to the left treap child of the Curr. Node
 8:     else
 9:         Set next node to the right treap child of the Curr. Node
10:     end if
11:     return Recursively on the Next Node
12: end function
```

Figure 4: Collection of path priority from treap

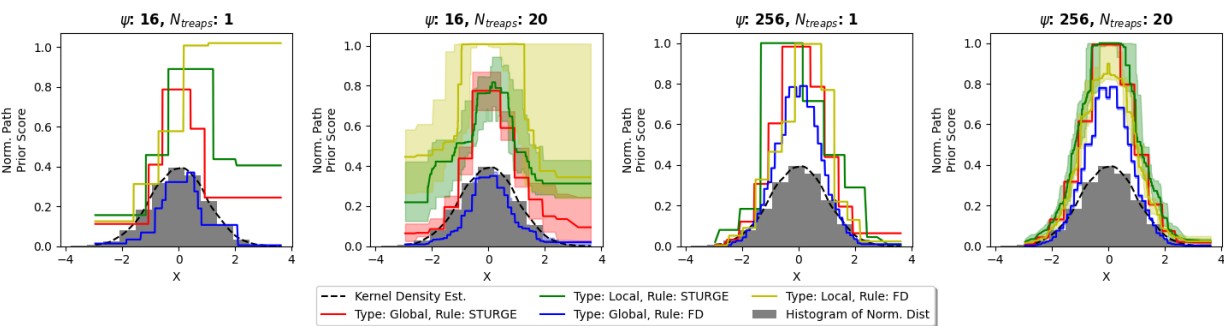

Figure 5: Yose-Ue normalized $p_{\mathrm{prior}}$ estimation given a univariate normal distribution. The dark lines represent the mean path traversal mass estimate score. The shaded regions represent the interquartile range of path-traversal mass estimates.

Yose-Ue constructs the treap ensemble through the training procedure outlined in Figure 3. Note that discretization will occur according to our estimation rule. In this work, we explore the effects of global mass estimation (deriving $B_{\mathrm{array}}$ and $B_{\mathrm{count}}$ based on $\mathcal{X}$) versus local mass estimation (constructing $B_{\mathrm{array}}$ and $B_{\mathrm{count}}$ based on $X_\psi$) for treap construction. Local (global) estimation enables greater diversity (uniformity) in split-point candidates.

**Performing Inference:** Yose-Ue traverses all treaps in the ensemble during inference. The treap traversal procedure is described in Figure 4. For each treap, a priority value ($p_{\mathrm{prior}}$) is incremented by the node's mass until no split nodes are available. We visualize our normalized $p_{\mathrm{prior}}$ scores (Figure 5) by estimating the mass of a normal distribution. Note that parameter $\psi$, the discretization function with estimation type, will indicate which aspects of our distribution will be mis-estimated, estimated efficiently, or over-estimated. These path scores are aggregated across all trees ($s_{\mathrm{paths}}$), as shown in Figure 6. The expectation of these scores ($p_{\mathrm{score}}$) is normalized according to the estimation rule. This normalized probability, $p_{\mathrm{prob}}$, is used as our input for our anomaly scoring statistic that follows $s = 2 - 2^{p_{\mathrm{prob}}}$.

```
 1: function predict(Yose-Ue, x_test, ψ)
 2:     s_paths ⇐ Collect mass estimates on all treaps in Yose-Ue ensemble for the x_test sample
 3:     if Est. Rule is Global then
 4:         p_prob ⇐ E[s_paths]/|X|
 5:     else
 6:         p_prob ⇐ E[s_paths]/ψ
 7:     end if
 8:     s = 2 − 2^{p_prob}
 9:     return s
10: end function
```

Figure 6: Yose-Ue Anomaly Scoring Function

### 4.5 Interpretable Mass Estimation for Anomaly Scoring

During treap traversal, the summed mass count $p_{prior}$ inherently gathers joint interactions as feature mass. Therefore, $p_{prior}$ implies that the proposed anomaly scoring mechanism provides insight into whether a test point $x_{test}$ originates from a multivariate region of high mass (nominal) or low mass (anomaly). Note that our final anomaly score $s$ is $2 - 2^{p_{prob}}$. Given that $p_{prob}$ is normalized, this implies that $p_{prob}$ is constrained to be between 0 and 1. The constraint on $p_{prob}$ imposes a constraint on the anomaly score, i.e., when $p_{prob}$ is 1 (0), the estimated mass is high (low), so the anomaly score is 0 (1). Therefore, by constructing our Yose-Ue mass estimator, we provide a direct, interpretable, and unsupervised method for anomaly detection.

Note that under a fixed discretization, empirical bin counts converge to their population bin probabilities as the sample size increases. Therefore, Yose-Ue can be interpreted as a proxy for an empirical mass model of the feature space, where finite partitions approximate relative feature mass for anomaly detection. In this view, the bin-level mass estimates used to guide the treap construction converge to their corresponding discretized population quantities, and the ensemble-averaged traversal score reflects relative mass across an ensemble of partition-based structures. This does not establish convergence to the true continuous density. Rather, we clarify that Yose-Ue estimates relative mass over a finite discretization of the feature space, instead of performing full continuous density estimation. A more rigorous treatment of score concentration, stability under subsampling, and discretization rules remains an important direction for future work.

A core component of Yose-Ue is its discretization-based mass estimation, which transforms input features into fixed-width histograms that underpin the empirical mass distribution of a given dataset ($\mathcal{X}$). This estimation can be performed globally (on $\mathcal{X}$) or locally ($X_\psi$). Global mass estimation provides uniformity in the split-priority landscape, enabling estimators to share a common structural understanding of feature-wise mass. Local mass estimation introduces adaptability, in which each tree independently constructs mass-estimation regions within its observation space. This estimation-type distinction provides a flexible tradeoff. Global mass leads to more uniform tree behavior and reduced variance, while local mass enhances expressiveness and can capture subtle, data-specific anomalies. We explore both of these estimation methods in Section 7.

## 5 Complexity Analysis

Note that a treap is essentially a randomized BST with a heap-style priority enforced during split selection and growth (Seidel & Aragon, 1996; Martínez & Roura, 1998). Treaps may be considered a type of randomized BST and have been shown to hold their essential construction and traversal properties (Seidel & Aragon, 1996). A randomized BST is structured as a proper binary tree, with every internal node having two children (Liu et al., 2008; Seidel & Aragon, 1996). In addition, note that a randomized BST is a balanced BST in terms of its expectation Martínez & Roura (1998); Seidel & Aragon (1996). We begin by presenting several lemmas that define the average-case training and inference complexities of a randomized BST ensemble (Martínez & Roura, 1998; Liu et al., 2008). We use these lemmas to establish the worst-case training and average-case inference complexity of the proposed treap ensemble.

### 5.1 Randomized BST Ensembles

We begin with the average-case training complexity of a randomized BST ensemble.

**Lemma 1.** *The average-case time complexity for training a randomized BST ensemble is $\Theta(N_{tree}\psi \log \psi)$, where $N_{tree}$ is the number of BSTs and $\psi$ is the subsampling size per tree.*

*Proof.* The average-case complexity to construct a randomized BST is $\psi \log \psi$ according to Stirling's approximation (Martínez & Roura, 1998; Felicioni, Nicolo, 2021). The complexity grows with the number of random BST estimators considered, i.e., $N_{tree}$. The complexity of training a randomized BST ensemble is $\Theta(N_{tree}\psi \log \psi)$ (Liu et al., 2008). □

Note that the average-case training of a BST ensemble grows linearly with the number of estimators $N_{tree}$ and grows log-linearly with the subsampling rate $\psi$. We next examine the average-case inference complexity of a BST ensemble.

**Lemma 2.** *The average-case search (inference) complexity is $\Theta(N_{tree} \log \psi)$.*

*Proof.* The average height of a random BST is $\log \psi$. Inference consists of a root-to-leaf traversal for all random BSTs. The average random BST root-to-leaf traversal length is considered to be the average random BST height (Martínez & Roura, 1998). Therefore, the average inference complexity for the random BST ensemble with $N_{tree}$ estimators is $\Theta(N_{tree} \log \psi)$. □

These lemmas establish the average-case complexity of the randomized BST ensemble. Note that the proposed treap ensemble is a modification of the randomized BST ensemble. Therefore, we use these BST-specific lemmas to establish a worst-case training and average-case inference complexity of treap ensembles.

## 5.2 Treap Ensembles

The set of candidate split points used to construct a treap is not determined solely by the $\psi$ sampled instances. Instead, it is upper-bounded by the size of the bin arrays ($B_{\text{array}}$, $B_{\text{count}}$), which are generated during the discretization procedure. The size of these arrays is governed by the number of bins considered during discretization, denoted by $\alpha$. The introduction of discretization and priority-based split-point selection directly influences both the training and inference complexities of the proposed treap ensembles. We formalize this impact through the following theorem and accompanying proof. We begin by analyzing the training complexity of the proposed treap ensemble.

**Theorem 1.** *Assuming that $\alpha < \psi$, Yose-Ue has a worst-case training complexity of $\mathcal{O}(N_{\text{treaps}}(\alpha \log \alpha + \alpha^2) + d)$ when the estimation type is global. Under the same assumption, but with estimation type local, the worst-case training complexity is $\mathcal{O}(N_{\text{treaps}}(\alpha \log \alpha + \alpha^2 + d))$.*

*Proof.* The cost of constructing a treap ensemble scales linearly with the number of treap estimators considered, i.e., $N_{\text{treap}}$. The proposed treap growth procedure is a modified random BST estimator with priority-based split points. Therefore, the cost of constructing a single treap estimator scales with the number of potential split points ($\alpha$) according to its underlying BST structure ($\alpha \log \alpha$) as per Lemma 1. However, the priority-based selection appends subsequent training cost per treap estimator. We begin by considering the additional training cost of priority-based split-point selection, and then provide insights into the cost of discretization.

In the construction of the proposed treap structure, split-point candidates are selected via an $\arg\max$ priority rule to preserve the heap properties of the treap. The cost of this split-point selection is the number of available split candidates. Note that the maximum number of split point candidates $\alpha$ arises at the beginning of treap construction. As the treap construction step is repeated, the available split-point space decreases. The split point candidacy set shrinks at the worst-case rate of $\alpha$, $\alpha - 1$, $\alpha - 2$, $\alpha - 3$, etc. However, note that this per-split point selection cost per treap simplifies to the following bound.

$$\alpha + (\alpha - 1) + (\alpha - 2) + \ldots = \sum_{i=0}^{\alpha - 1} (\alpha - i)$$
$$= \frac{\alpha(\alpha + 1)}{2}$$
$$\leq \alpha^2 \tag{13}$$

Therefore, the arg max priority-value selection procedure incurs an additional $\mathcal{O}(\alpha^2)$ cost during treap construction. In conjunction, the cost of discretization scales linearly with the number of dimensions/features ($d$) in our original dataset $\mathcal{X} \subset \mathbb{R}^d$. In this study, our discretization functions can be computed directly for each feature. Hence, the discretization-induced complexity per feature is $\mathcal{O}(1)$, which indicates the total complexity for the discretization procedure is $\mathcal{O}(d)$. Finally, depending on whether discretization is used locally ($X_\psi$) or globally ($\mathcal{X}$), the cost of discretization is either incurred once (globally) or for each treap estimator (locally). Therefore, assuming that $\alpha < \psi$, the worst-case training complexity of Yose-Ue when estimating globally is $\mathcal{O}(N_{\text{treaps}}(\alpha \log \alpha + \alpha^2) + d)$. When Yose-Ue is estimating locally, the worst-case training complexity is $\mathcal{O}(N_{\text{treaps}}(\alpha \log \alpha + \alpha^2 + d))$ $\qquad \square$

Depending on whether $\alpha < \psi$ and the data dimensionality $d$, the training complexity of Yose-Ue may be higher than that of classical BST ensembles. However, this depends on the choice of the discretization function, estimation rule, and the dataset's dimensionality. As a result of Theorem 1, we construct the following corollary to derive this tradeoff as an inequality.

**Corollary 1.** *The Yose-Ue training cost, with locally estimated discretization, is lower than that of an equally-sized randomized BST ensemble when the inequality $\alpha 2^{(\alpha d)/\psi} < \psi^{\psi/\alpha}$ is true. When globally-driven discretization is employed under the same assumption, Yose-Ue has lower training complexity when $\alpha 2^{(\alpha d)/(\psi N_{\text{treaps}})} < \psi^{\psi/\alpha}$.*

*Proof.* Note that as each ensemble uses the same number of estimators, we must ensure that the complexity of constructing a single treap estimator is lower than that of a randomized BST estimator. We begin with the scenario in which Yose-Ue uses locally estimated discretization. In addition, we assume the total number of candidates from discretization ($\alpha$) is a ratio ($\epsilon$) of $\psi$, i.e., $\alpha = \epsilon\psi$ where $\epsilon \in [0, 1]$. We begin with the inequality

$$\alpha \log \alpha + \alpha^2 + d < \psi \log \psi.$$

Let $\alpha = \epsilon\psi$. Substituting this into the inequality implies

$$\epsilon\psi \log(\epsilon\psi) + (\epsilon\psi)^2 + d < \psi \log \psi.$$

Expanding the logarithm using $\log(\epsilon\psi) = \log \epsilon + \log \psi$ gives

$$\epsilon\psi(\log \epsilon + \log \psi) + \epsilon^2\psi^2 + d < \psi \log \psi.$$

Dividing both sides by $\psi > 0$, we obtain

$$\epsilon(\log \epsilon + \log \psi) + \epsilon^2\psi + \frac{d}{\psi} < \log \psi.$$

Rearranging terms therefore yields

$$\log(\epsilon^\epsilon) + \log(\psi^\epsilon) + \epsilon^2\psi < \log \psi - \frac{d}{\psi}.$$

We then exponentiate both sides with base 2, which implies

$$(\epsilon^\epsilon)(\psi^\epsilon)2^{(\epsilon^2\psi)} < \psi 2^{-\frac{d}{\psi}}.$$

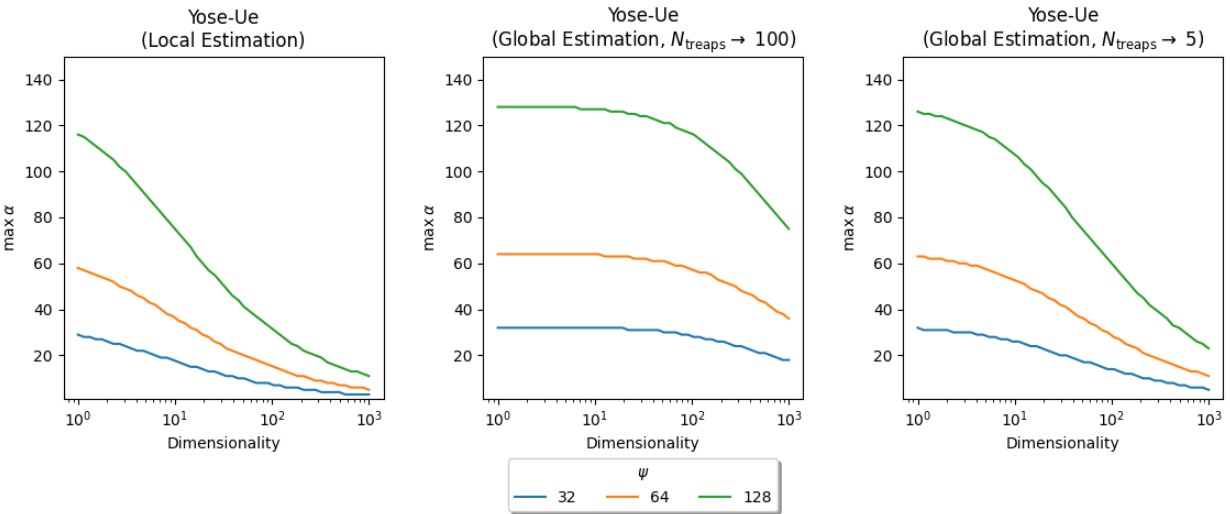

Figure 7: The training complexity tradeoff boundary for the proposed treap ensembles compared with randomized BST ensembles. The lines indicate the maximum number of split-point candidates permissible (for a given data dimensionality) before treap ensemble construction becomes more expensive than a randomized BST ensemble.

Consequently,

$$\epsilon\psi 2^{\epsilon d} < \psi^{1/\epsilon}.$$

Substituting back $\alpha = \epsilon\psi$ therefore gives the equivalent form

$$\alpha 2^{(\alpha d)/\psi} < \psi^{\psi/\alpha}.$$

A similar derivation applies to the scenario in which Yose-Ue uses a globally estimated discretization. In this scenario, the cost of discretization per treap is $d/N_{\text{treaps}}$, and the inequality is as follows.

$$\alpha \log \alpha + \alpha^2 + \frac{d}{N_{\text{treaps}}} < \psi \log \psi$$

By rearranging and exponentiating the inequality (with base 2), we obtain the equivalent condition

$$\alpha 2^{(\alpha d)/(\psi N_{\text{treaps}})} < \psi^{\psi/\alpha}.$$

Hence, the preceding transformations show that the original logarithmic inequality implies the stated exponential bound. Thus, we establish the desired result and complete the proof.

$\square$

As shown in Theorem 1 and Corollary 1, by selecting an appropriate discretization function and estimation rule for a dataset of dimension $d$, the proposed treap ensemble can improve training efficiency over randomized BST ensembles. We visualize the training tradeoff inequality boundary in Figure 7. The plots show the maximum number of split-point candidates ($\alpha$) permissible for various subsampling rates ($\psi$) and data dimensions. Following our derivation of the worst-case training complexity of Yose-Ue, we now provide the average-case inference complexity.

**Theorem 2.** *Assuming $\alpha \leq \psi$, the average-case inference complexity of Yose-Ue is $\Theta(N_{\text{treaps}} \log \alpha)$.*

*Proof.* The underlying structure of the treap is a BST. A BST has an average search cost of $\log \psi$ as per Lemma 2. However, since there are $\alpha$ split-points to isolate rather than $\psi$ samples, the average search cost per treap is $\log \alpha$. This search cost scales linearly with the total number of treap estimators in the proposed ensemble method. The average-case inference complexity for Yose-Ue is $\Theta(N_{\text{treaps}} \log \alpha)$ □

Depending on the values of $\alpha$ and $\psi$, the inference cost of the proposed ensemble is lower than that of BST ensembles. However, even when $\alpha > \psi$, we can construct the following corollary.

**Corollary 2.** *When $\alpha > \psi$, the average-case inference complexity of Yose-Ue is that of an equivalently sized BST ensemble, i.e., $\Theta(N_{\text{treap}} \log \psi)$.*

*Proof.* Theorem 2 states that the average-case inference complexity of the proposed ensemble is $\Theta(N_{\text{treap}} \log \alpha)$ due to the isolation of the split-candidates $\alpha$ when $\alpha \le \psi$. However, the training procedure of the proposed treap structure contains the original termination conditions of the randomized BST (Liu et al., 2008). Therefore, even when $\alpha > \psi$, the total number of isolated points will be bounded by the subsampling size $\psi$ or rather $\min(\psi, \alpha)$. This implies that the average height and the subsequent average-case search cost of an estimator are bounded by $\log \psi$ when $\alpha > \psi$. Therefore, the complexity of the average-case inference complexity of the treap ensemble when $\alpha > \psi$ is $\Theta(N_{\text{treap}} \log \psi)$. □

An implication of Theorem 2 and Corollary 2 is that the average-case inference complexity of the proposed method is either equal to or lower than a BST ensemble, regardless of the values of $\alpha$ and $\psi$. Now that we have established the computational complexity of the proposed method, we next provide insights into the estimation-type hyperparameter, i.e., global versus local.

# 6 Datasets & Experimental Setup

We evaluate Yose-Ue on 14 datasets. These consist of three synthetic datasets widely used in the anomaly detection literature (Ortega et al., 2025; Liu et al., 2008; Hariri et al., 2021), six datasets from a popular real-world anomaly detection benchmark (Rayana, 2016), two of these ODDS datasets with kurtosis dimensionality reduction, and the last three datasets originate from anomaly use-cases for experimental iPhone sensor data (Ortega et al., 2025; Ortega, Eduardo, 2025). A breakdown of these datasets and their acronyms used in this work is presented in Table 2. Note that $|\mathcal{X}|$ represents the number of normal points, $d$ signifies the number of dimensions in $\mathcal{X}$ for a given dataset, $X_{\text{ano}}$ represents the anomaly-specific validation set, and $|X_{\text{ano}}|$ denotes the number of anomaly data points available for testing.

The data set $\mathcal{X}$ is partitioned into a train- and test-set ($X_{\text{train}}/X_{\text{nom}}$). The train set, $X_{\text{train}}$, is used to construct our treap ensemble during training. The test set, $X_{\text{nom}}$, is used as our nominal validation set. Our exploration of Yose-Ue's design space considers train/test splits of 95/05, 90/10, 80/20, 70/30, and 60/40. Aside from our nominal training/testing splits, no initial seeds were used in this work; thus, all results within a specific train/test split are inherently cross-seed validation. After training on $X_{\text{train}}$, the subsequent anomaly scores of $X_{\text{nom}}$ and $X_{\text{ano}}$ are used to compute the AUCROC and AUCPR, which are popular anomaly detection performance metrics (Xiao & Fan, 2024; Seits et al., 2022; McDermott et al., 2024). Our experiments are conducted on an Intel Core i5-12600K CPU running Ubuntu 20.04.6 LTS with Kernel 5.15.0-101-generic. Each method is benchmarked 10 times per dataset. During our benchmarking, we extract the mean and standard deviation of the training/test times, memory usage, and performance metrics (AUCPR, AUCROC).

# 7 Design Space Exploration

The AUCPR and AUCROC results are shown in Figure 8 and Figure 9, respectively. The input parameters $N_{treaps}$ and $\psi$ are swept for each subplot. The number of estimators ($N_{treaps}$) is swept from $1, 5, 10, 15, ..., 100$. The subsampling size ($\psi$) is swept from 16, 32, 64, 128, 256 to 512. For each figure, the leftmost column signifies Yose-Ue trained and tested using Sturge's rule with global estimation. The second left-most column indicates the proposed treap ensemble trained using the global estimation type with Freedman Diaconis

Table 2: Datasets to evaluate the proposed method.

| Dataset | Abbreviation | $|\mathcal{X}|$ | $d$ | $|X_{\text{ano}}|$ |
|---|---|---|---|---|
| **ONE CLUSTER** | C1 | 1000 | 2 | 100 |
| **TWO CLUSTER** | C2 | 2000 | 2 | 100 |
| **SINUSOIDAL** | SIN | 1000 | 2 | 100 |
| **FOREST COVER (Rayana, 2016)** | FC | 286048 | 10 | 2747 |
| **SMTP (Rayana, 2016)** | SMTP | 95156 | 3 | 30 |
| **PENDIGITS (Rayana, 2016)** | PEN | 6870 | 16 | 156 |
| **THYROID (Rayana, 2016)** | THY | 3772 | 6 | 93 |
| **SATELLITE (Rayana, 2016)** | ST | 4435 | 36 | 672 |
| **SATIMAGE-2 (Rayana, 2016)** | ST-2 | 5803 | 36 | 71 |
| **FOREST COVER (Rayana, 2016) + Kurtosis** | FC-K | 286048 | 6 | 2747 |
| **SATELLITE (Rayana, 2016) + Kurtosis** | ST-K | 4435 | 9 | 672 |
| **EDGE | Car Crash** (Ortega et al., 2025) | E-CC | 888 | 6 | 203 |
| **EDGE | Magnetic Fluctuations** (Ortega et al., 2025) | E-MF | 710 | 3 | 100 |
| **EDGE | Vibrations** (Ortega et al., 2025) | E-VI | 666 | 6 | 100 |

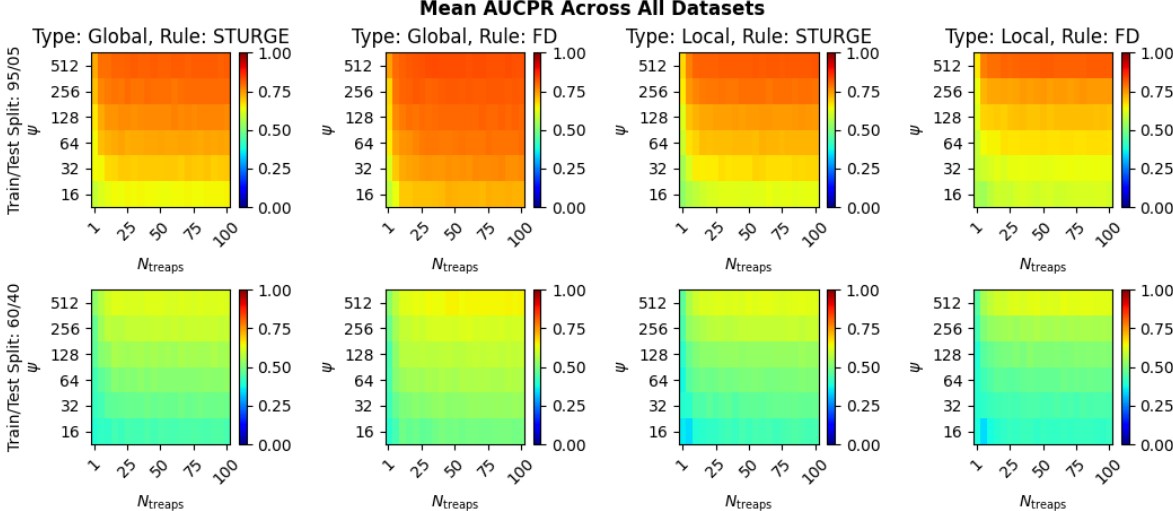

Figure 8: Performance precision-recall design space exploration for the Yose-Ue treap ensemble method. All subplots show the mean AUCPR across all datasets.

discretization. The subsequent columns use a local estimation rule, first with Sturge (the second right-most column) and then with Freedman Diaconis discretization (the right-most column), to train Yose-Ue. For brevity, we present the mean performance for the training/testing splits of 95%/05% (top row) and 60%/40% (bottom row). The best-performing Yose-Ue instances are obtained with the Freedman Diaconis/Sturge discretization function, applied globally/locally.

To understand the trade-offs between these two iterations of Yose-Ue, we present their performance and memory curves on the FC dataset across different train/test splits. We consider $\psi$ for 16 and 512 with 1, 5, 20, 40, 80, and 100 estimators. The results are shown in Figure 10. The points (error bars) on the performance-memory curve represent the mean (standard deviation). It is clear that Yose-Ue with Freedman Diaconis discretization applied globally achieves better performance than the Yose-Ue with locally estimated Freedman Diaconis discretization. However, we observe that Yose-Ue with locally estimated Sturge discretization has improved memory efficiency. Note that the locally estimated discretization procedure enables further data variation to be captured per treap estimator. This results in greater fidelity in data representation and, empirically, reduces the memory footprint per treap estimator.

Yose-Ue performs most effectively on datasets where the nominal distribution exhibits a stable mass structure that can be captured through discretized partitioning, including heterogeneous and moderately multimodal densities (e.g., C1, C2). In such cases, the mass-aware treap construction can better represent local geometry and yield stable anomaly scores. Conversely, datasets characterized by extremely sharp or rapidly varying density regions may require finer discretization to fully resolve underlying structure (e.g., ST-K, FC-K). These distributional characteristics help explain the observed empirical trends: datasets with well-separated

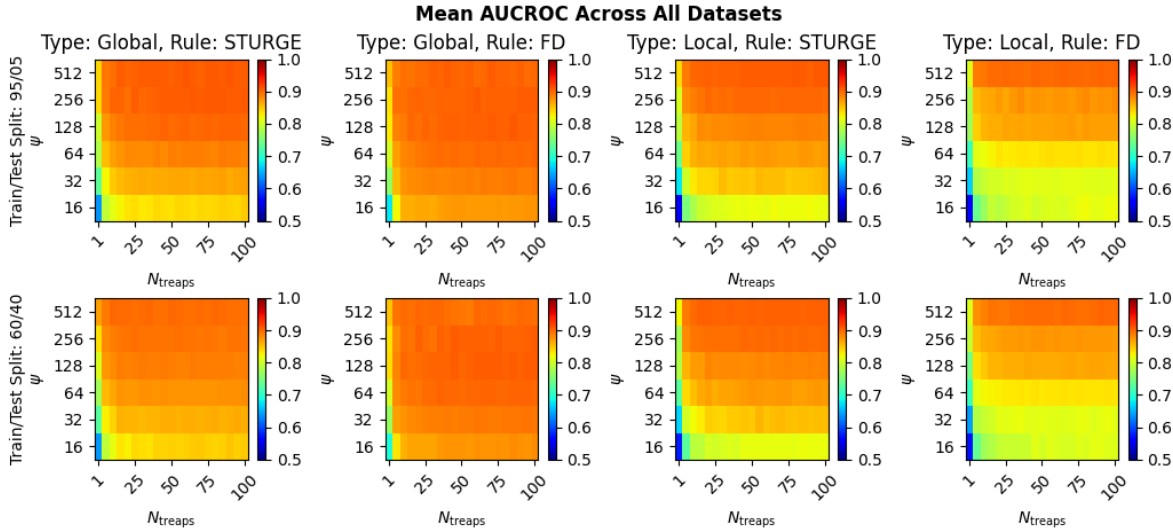

Figure 9: The AUCROC performance design space exploration for the Yose-Ue treap ensemble method. All subplots show the mean AUCROC across all datasets.

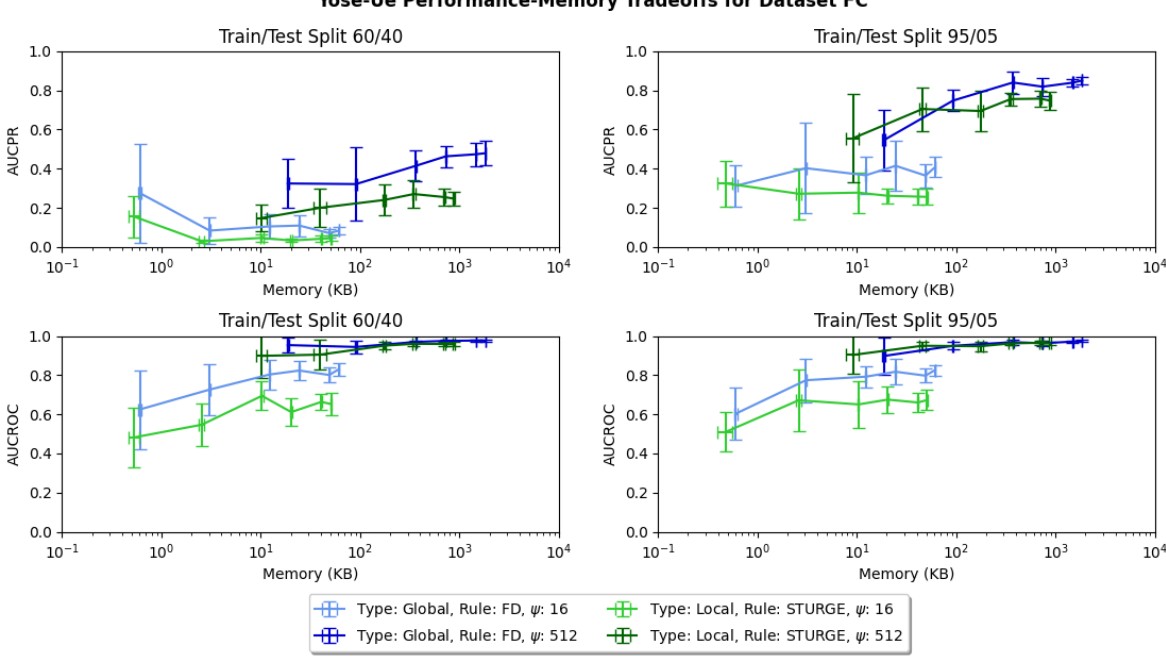

Figure 10: Performance–Memory Tradeoff for Yose-Ue design space exploration for the Forest Cover dataset (FC). The number of treap estimators considered is 1, 5, 20, 40, 80, and 100.

mass structure exhibit consistent performance improvements, whereas those with high-frequency density variation show increased sensitivity to discretization granularity. Note that this characterization should be interpreted as an empirical guideline rather than a formal distributional guarantee. However, these results highlight the importance of selecting appropriate discretization strategies when applying Yose-Ue in practice. In this work, we adopt Sturge's rule for local discretization to provide stable local partitioning. The Freedman Diacnois' rule is employed for global discretization, enabling finer data-adaptive resolution across the feature space.

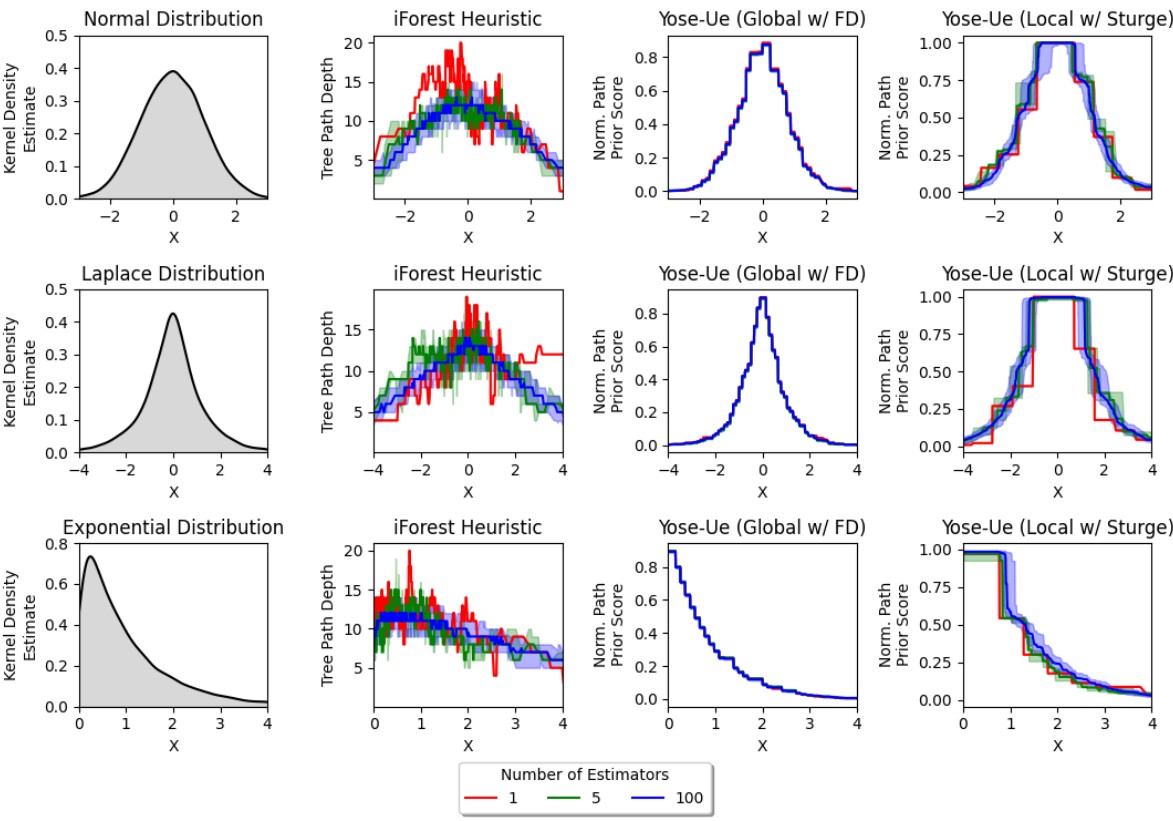

Figure 11: Distributional representation study from the BST ensemble path-depth estimates compared with the proposed treap ensemble path-traversal mass estimates. The dark lines are the median estimate, and the shaded region represents the interquartile range of the estimates.

## 8 Improved Distribution Representation

A key contribution of this paper is the mass-estimated path traversal procedure for the proposed mass-informed treap estimators. These mass-informed estimates align more closely with nominal data representations than SOTA path-depth estimates (Liu et al., 2008). Therefore, the considered baseline is the prior SOTA path-depth heuristic, widely used in many iForest variants (Liu et al., 2008; Hariri et al., 2021; Xu et al., 2023; Ortega et al., 2025). To evaluate the impact of the proposed path-traversal mass estimator versus the original path-depth heuristic, we visually compare a kernel density estimator (KDE) with each of these treap/tree-based estimators across several known distributions (Gaussian, Laplace, and Exponential). The KDE estimator with Scott's rule is used in our distribution estimation comparison. We consider a univariate Gaussian (top row), Laplace (middle row), and exponential (bottom row) distributions. For fair comparison, we use a subsampling rate of 256, as in the original iForest method (Liu et al., 2008). We sweep the number of estimators to be 1, 5, and 100 for brevity. The results are shown in Figure 11. The leftmost column shows the original distribution estimate with KDE. The following columns present the variant estimates obtained from the original iForest path-depth estimation and the mass estimate via Yose-Ue. Based on this design space exploration, we use Yose-Ue with the Global rule and Freedman Diaconis discretization, and the local rule with Sturge's rule, for our mass estimator ablation study.

We observe limited diversity in mass estimates when either very few or many treap ensemble estimators are used under the global rule with Freedman Diaconis discretization. In contrast, increasing the number of estimators under the local rule with Sturge's discretization produces progressively smoother mass estimates. Nevertheless, even with only five estimators – and across all Yose-Ue hyperparameter configurations (global with Freedman Diaconis or local with Sturges) – the proposed method accurately captures the nominal

Table 3: Kolmogorov-Smirnov (KS) statistic of KDE distribution estimation compared with ensemble method estimation. A higher KS statistic indicates greater misalignment between the Gaussian KDE with the ensemble distribution estimate.

| Distribution | Number of Estimators | iForest | Yose-Ue (Global, FD) | Yose-Ue (Local, Sturge) | KS Statistic Reduction |
|---|---|---|---|---|---|
| $\mathcal{N}(0,1)$ | 1 | 0.234 | 0.054 | 0.023 | 4.33–10.17× |
| | 5 | 0.199 | 0.058 | 0.033 | 3.43–6.03× |
| | 100 | 0.173 | 0.055 | 0.023 | 3.14–7.52× |
| $\mathcal{L}(0,1)$ | 1 | 0.337 | 0.058 | 0.142 | 2.37–5.81× |
| | 5 | 0.221 | 0.062 | 0.100 | 2.21–3.56× |
| | 100 | 0.216 | 0.061 | 0.071 | 3.04–3.54× |
| $\mathrm{Exp}(1)$ | 1 | 0.379 | 0.106 | 0.048 | 3.57–7.89× |
| | 5 | 0.394 | 0.095 | 0.039 | 2.43–4.14× |
| | 100 | 0.390 | 0.101 | 0.055 | 3.86–7.09× |

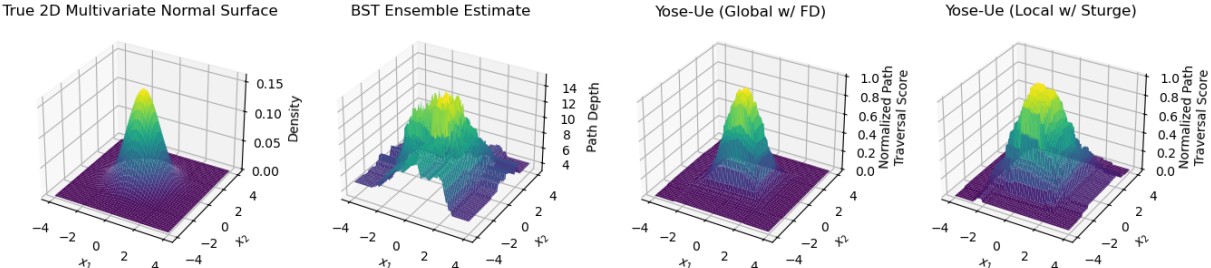

Figure 12: True multivariate normal density, BST-ensemble estimate, and proposed treap mass estimate surfaces.

density across the considered distributions. In comparison, when density is approximated using the path-depth heuristic (as in iForest), the resulting estimates struggle to recover the underlying distributional structure and improve only when a large number of estimators is employed. To quantify these observations, we compute the Kolmogorov-Smirnov (KS) statistic for each ensemble-based density estimate relative to a Gaussian KDE reference across all distributions. The resulting KS statistics are reported in Table 3. Larger KS values indicate greater discrepancy between the KDE and the ensemble estimate. As shown, the path-depth estimator (iForest) consistently produces higher KS statistics than Yose-Ue. Although the KS statistic of the path-depth approach decreases as the ensemble size grows, Yose-Ue maintains strong alignment with the nominal distribution even with a single estimator. Overall, Yose-Ue reduces the KS statistic by up to an order of magnitude (10×) relative to state-of-the-art path-depth estimation methods, i.e., iForest. Note that the KS statistic is used as an empirical diagnostic for distributional alignment with a reference density estimate. In future work, finite-sample confidence bounds may be constructed for treap-learners to determine whether the mass-traversal score converges to the true density.

Note that as Yose-Ue is aimed to provide a multivariate mass estimate, we also consider the multivariate scenario for our distribution-representation baseline analysis. In this scenario, we consider a multivariate normal distribution, i.e., $\mathcal{N}(\vec{0}, \mathbf{I}_d)$ where $\vec{0}$ is the zero-mean vector and $\mathbf{I}_d$ is the $d \times d$ identity covariance matrix. We compare the original multivariate distribution with the BST path-depth estimates and the treap-based path traversal scores. The number of estimators considered in this scenario is 5. The multivariate distribution and ensemble-based estimates are visualized as 3D surface plots in Figure 12. As shown, the proposed treap-learner framework better encapsulates the multivariate distribution representation than the path-depth BST ensemble techniques.

In resource-constrained deployment settings, only a limited number of estimators may be feasible. Under such constraints, it is critical that the ensemble provides a well-calibrated representation of the nominal distribution even with a few estimators. In iForest, anomaly scoring is driven by BST path-depth estimation. However, when only a small number of trees are used, this heuristic struggles to preserve the underlying density ordering of the data, e.g., a few BSTs. Preserving the nominal distribution is particularly important in unsupervised anomaly detection, where notions of density, proximity, and isolation are defined relative to the

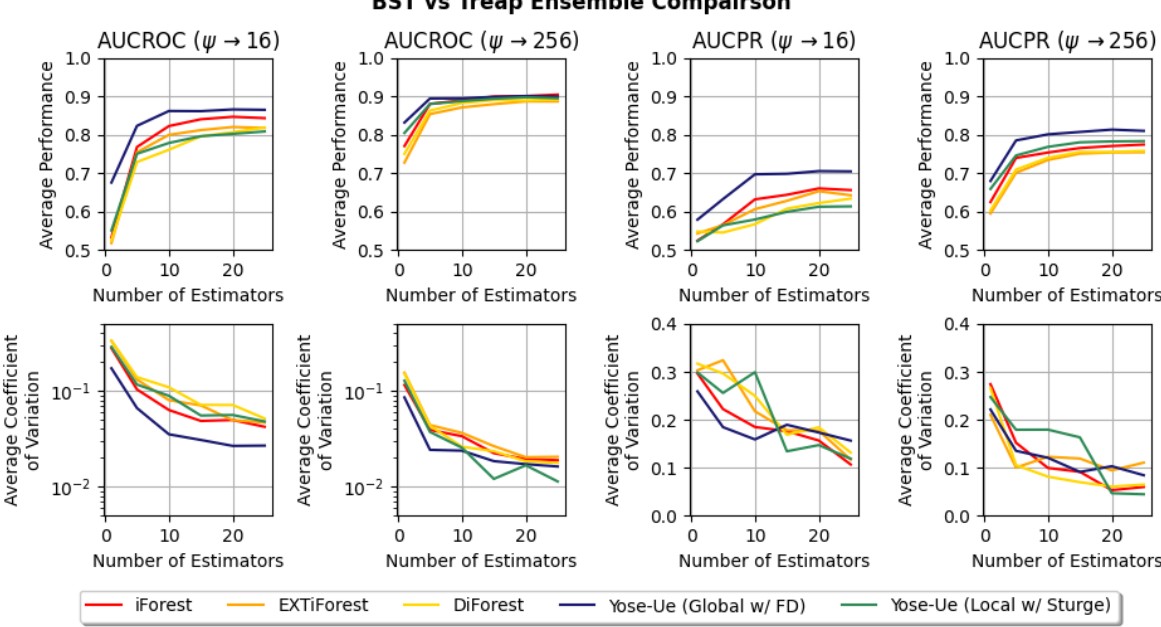

Figure 13: The average anomaly detection performance (AUCROC, AUCPR) and average coefficient of variation of iForest-based (BST ensembles) compared with Yose-Ue (treap ensemble). The training/testing split is 95/5.

structure of nominal data. If the ensemble distorts this structure, anomaly scores may no longer meaningfully reflect true deviations. In contrast, the proposed treap mass-estimate scoring mechanism better preserves the nominal distribution, as evident from Figure 11 and Table 3. Having established that the proposed scoring approach improves distributional fidelity, we next demonstrate that this improved nominal representation translates into enhanced anomaly detection performance for ensemble-based methods.

## 9 BST vs Treap Ensembles

Figure 13 reports the average anomaly detection performance with the average detector coefficient of variation (CV). We consider a deployment regime in which the inference system is restricted to a small ensemble size, specifically $1, 5, 10, 15, 20, 25$ estimators per method (iForest, EXTiForest, and DiForest) across all datasets. The top row presents the mean detection performance, while the bottom row reports the corresponding coefficient of variation, capturing estimator stability. The two left columns show AUCROC results, and the two right columns show AUCPR results. To evaluate the effect of sample size on estimator diversity, we vary the subsampling rate $\psi \in 16, 256$. The value $\psi = 256$ is included for consistency with the original iForest formulation (Liu et al., 2008). As shown, the proposed treap ensemble with global mass estimates consistently outperforms BST-based ensemble methods in low-estimator regimes. In particular, Yose-Ue with global mass estimates exhibits lower relative AUCROC variance and improved average AUCPR performance when $\psi$ is small. Furthermore, the treap ensemble with local mass estimates performs competitively when $\psi$ is larger. Notably, increasing $\psi$ enables local treap estimators to incorporate more information during construction, thereby further stabilizing detection performance.

To further evaluate ensemble behavior under resource constraints, we construct Gaussian mixture contour plots of joint performance (AUCROC, AUCPR) across all benchmarks. Each benchmark contributes its mean performance along with its relative deviation[2]. In this experiment, we restrict the ensemble size to five estimators to reflect a resource-constrained deployment setting. The average memory usage for

---

[2]A Gaussian KDE estimator is used to construct the bivariate performance surface. 15 contour levels are used, with a bandwidth of 0.7.

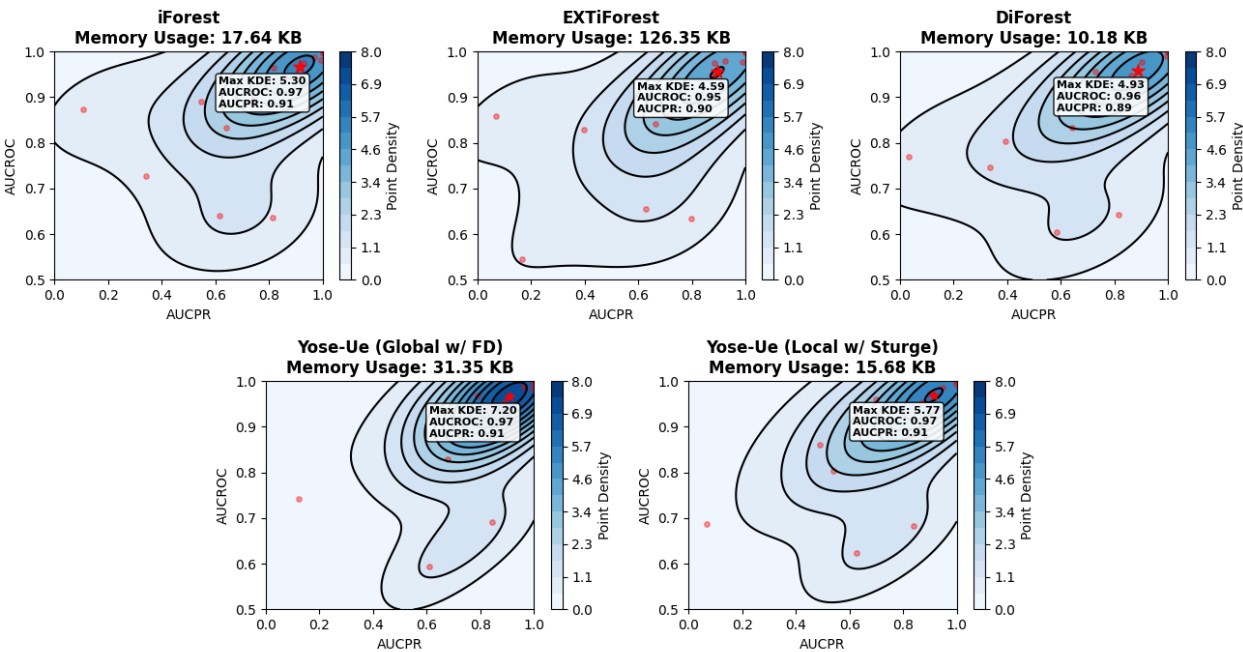

Figure 14: BST-ensemble compared with treap-based approach for anomaly detection performance (AU-CROC, AUCPR) under resource-constrained settings (only five estimators in ensemble). The subsampling rate ($\psi$) is 256, and the training/testing split is 95/5.

Table 4: Average performance improvement for the Yose-Ue treap Learning method compared with prior BST ensemble methods. The subsampling size $\psi$ is 256, and the number of estimators is 5. Both AUCPR and AUCROC are reported as improvements for training/testing splits of 95/5, 90/10, and 80/20.

| Train/Test Split | BST Ensembles | Yose-Ue (Global with FD) | | | Yose-Ue (Local with Sturge) | | |
|---|---|---|---|---|---|---|---|
| | | $\Delta$AUCROC | $\Delta$AUCPR | $\Delta\mathbb{E}[d_b]$ | $\Delta$AUCROC | $\Delta$AUCPR | $\Delta\mathbb{E}[d_b]$ |
| 95/5 | iForest | 0.013 | 0.045 | 0.048 | -0.0003 | 0.006 | 0.007 |
| | EXTiForest | 0.040 | 0.084 | 0.094 | 0.026 | 0.045 | 0.053 |
| | DiForest | 0.031 | 0.075 | 0.082 | 0.017 | 0.036 | 0.041 |
| 90/10 | iForest | 0.021 | 0.061 | 0.065 | 0.007 | 0.026 | 0.027 |
| | EXTiForest | 0.025 | 0.068 | 0.072 | 0.010 | 0.032 | 0.033 |
| | DiForest | 0.032 | 0.064 | 0.071 | 0.017 | 0.029 | 0.033 |
| 80/20 | iForest | 0.005 | 0.032 | 0.032 | -0.007 | 0.002 | -0.002 |
| | EXTiForest | 0.031 | 0.066 | 0.071 | 0.017 | 0.036 | 0.038 |
| | DiForest | 0.023 | 0.059 | 0.063 | 0.010 | 0.029 | 0.030 |

each method is reported in the corresponding subplot title. The resulting bivariate KDE contour plots are shown in Figure 14. For each method, the mean benchmark performance is indicated by an opaque red dot. The red star marks the peak of the estimated performance density and is annotated with the corresponding peak density value and its associated AUCROC and AUCPR coordinates. All ensemble methods achieve comparable peak performance, with AUCROC ranging from 0.95 to 0.97 and AUCPR ranging from 0.89 to 0.91. However, the BST-based ensemble methods (iForest, DiForest, EXTiForest) exhibit a more dispersed joint performance distribution, resulting in lower maximum density concentrations. Specifically, their peak density estimates range from 4.93 to 5.30, whereas the proposed treap-based Yose-Ue methods attain higher peak densities of 5.77 (local) and 7.20 (global). This indicates that treap ensemble methods produce performance distributions that are more tightly concentrated in regions of simultaneously high AUCROC and AUCPR, whereas BST-based ensembles display greater variability across benchmarks. In other words, while peak performance is similar across methods, Yose-Ue achieves greater consistency in performance under constrained ensemble sizes.

To quantify the performance gains of treap ensembles over BST ensembles, we report the average anomaly detection improvements in Table 4 and introduce an aggregate distance-to-ideal performance metric. We

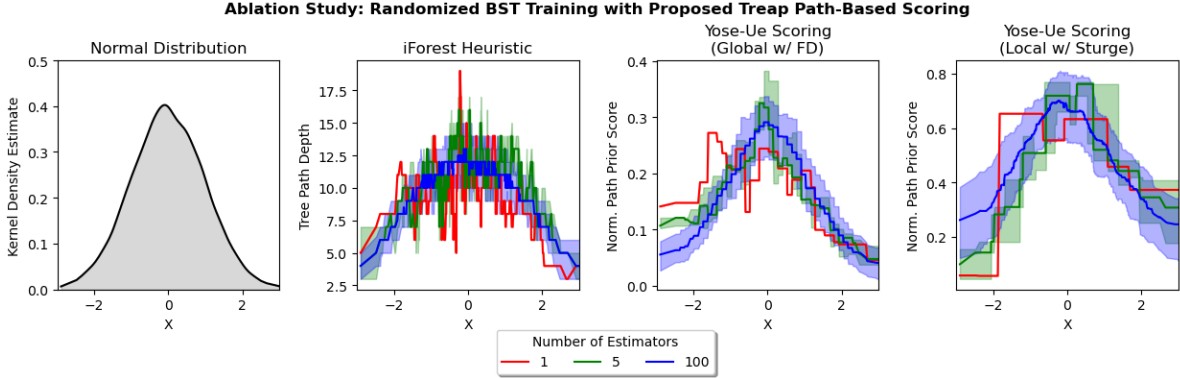

Figure 15: An ablation study of the BST ensemble method compared to a BST ensemble equipped with the proposed treap ensemble path-traversal mass estimates. The dark lines are the median estimate, and the shaded region represents the interquartile range of the estimates.

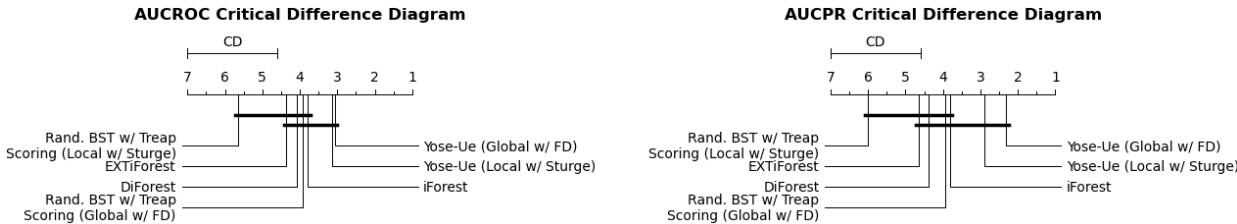

Figure 16: Critical difference diagram of forest-based methods for anomaly detection. The value of $\psi$ is 256, the training/testing split is 95/5, and the number of estimators considered is 5.

define the ideal anomaly detection point as $\vec{z}^* = \langle 1.0, 1.0 \rangle$, corresponding to perfect AUCPR and AUCROC. For each benchmark and method, we represent the expected performance as a point in the joint AUCPR-AUCROC space, i.e., $\vec{b} = \langle \mu_{\mathrm{AUCPR}}, \mu_{\mathrm{AUCROC}} \rangle$. We then compute the Euclidean distance to the ideal point represented as $d_b = \|\vec{z}^* - \vec{b}\|_2$. Each method yields a set of distances $d_b$ across all benchmarks. To measure improvement relative to BST-based baselines, we compute the expected $d_b$ reduction, i.e., $\Delta\mathbb{E}[d_b]$, which captures the average shift toward optimal anomaly detection performance. A larger value of $\Delta\mathbb{E}[d_b]$ indicates a greater improvement achieved by the proposed treap ensemble (Yose-Ue) relative to BST-base ensembles. As shown, across the considered train/test splits, the treap ensemble consistently reduces the distance to the ideal performance point and, on average, outperforms all selected BST ensemble baselines.

## 10 Contribution of Treap Construction and Path-Traversal Scoring

To further highlight the contributions of the proposed treap construction and path-traversal scoring components, we construct a hybrid randomized BST ensemble that utilizes the proposed treap traversal-based scoring strategy. Figure 15 illustrates the behavior of the scoring mechanism on a Normal distribution, while Figure 16 presents critical difference (CD) diagrams computed from the expected AUCROC and AUCPR rankings across all benchmark datasets compared with the proposed treap framework and SOTA forest-based methods benchmarked previously, i.e., DiForest, EXTiForest, and iForest.

The synthetic experiment highlights a key distinction between conventional isolation-based scoring and the proposed traversal-based mass estimate. Under the iForest heuristic, path depths exhibit substantial variability and require large ensembles to approximate the underlying data distribution. In contrast, the proposed path-traversal score produces a smoother empirical mass profile that more closely follows the shape of the underlying normal distribution (Figure 15). As the ensemble size increases, the traversal score converges toward a stable estimate of relative feature-space mass. However, when we compare the treap-

based estimation (Figure 11) with the modified randomized BST ensemble (Figure 15), the randomized BST variants exhibit a wider range of scoring behavior than the proposed treap ensemble. Therefore, this suggests that the treap-construction technique (with inverse-mass priority-based splits) helps reduce our observed mass-estimation uncertainty.

To determine whether these observations translate into improved anomaly detection performance, we evaluate the ablated variants against the state-of-the-art tree-ensemble baselines and the full treap models. We used average-rank and critical difference analyses based on the expected AUCROC ($\mu_{\text{AUCROC}}$) and the expected AUCPR ($\mu_{\text{AUCPR}}$) per dataset. Interestingly, randomized BST ensembles equipped with the proposed traversal score outperform DiForest, which is also based on discretized partitions of the feature space. This suggests that the proposed scoring mechanism captures meaningful information about the underlying distribution's shape, thereby improving anomaly detection performance. However, we observed that the randomized BST variants, without treap construction, did not achieve the same performance rankings as the original iForest or the full treap-based models. This indicates that while the scoring rule improves distributional interpretability (as shown in Figure 15), the treap construction improves estimator stability and therefore, contributes to the improved anomaly-detection performance.

In addition, we observed that across AUCROC and AUCPR, the proposed treap learner framework (Yose-Ue) achieves the best average ranks among all compared methods (Figure 16). In particular, the global Freedman–Diaconis discretization consistently attains the highest ranking, followed closely by the local Sturges-based variant. These results suggest that both components matter: the traversal score improves the information captured by standard randomized BST ensembles, while the inverse-mass treap construction improves estimator stability. The CD analysis supports a consistent ranking advantage for Yose-Ue, although not all close comparisons are statistically distinguishable.

## 11  Comparison with State-Of-The-Art Methods

To compare the proposed treap ensemble (Yose-Ue) against resource-constrained SOTA approaches, we consider both DNN and non-DNN methods. For the DNN baselines, we evaluate an autoencoder (AE) and a graph attention network autoencoder (GATE) (Chen et al., 2018; Salehi & Davulcu, 2020). To ensure a fair comparison in resource-constrained settings, both architectures are configured to have a memory footprint comparable to Yose-Ue. In addition, to ensure a fair comparison on resource-constrained anomaly detection, methods that require specific accelerator hardware are excluded from the comparison, e.g., an LLM Tsai et al. (2025). Note that our SOTA baseline comparisons are intended to evaluate Yose-Ue as a practical, interpretable, and memory-efficient alternative when deep models are not deployable, rather than as a universal replacement for deep anomaly detectors in unconstrained environments. The GATE model consists of five hidden layers, each with a dimensionality of 10. The AE architecture comprises eleven hidden layers with dimensions 64, 32, 16, 8, 4, 2, 4, 8, 16, 32, 64, forming a symmetric bottleneck structure. All DNN methods are trained for 50 epochs under identical training conditions. For resource-efficient non-DNN baselines, we include Histogram-Based Outlier Scores (HBOS) and Local Outlier Factor (LOF) (Aguilera-Martos et al., 2023; Breunig et al., 2000). Both methods are implemented using the default PyOD hyperparameters. We observed that both HBOS and LOF are inherently memory efficient with their default parameters. For LOF, the ball-tree algorithm is used to ensure computational consistency across datasets. In all comparisons against SOTA methods, Yose-Ue is restricted to five treap estimators to reflect a resource-constrained deployment regime. All results are reported in Table 5.

The performance of each SOTA method is further visualized using Gaussian KDE contour plots in Figure 17. Each subplot title shows the method acronym and its average inference-time memory usage across datasets. The GATE, AE, and HBOS methods exhibit performance densities concentrated near the saturation region of the AUCPR-AUCROC space, with a pronounced ridge structure. This pattern suggests that performance variability across benchmarks is largely confined to a single coupled direction in this joint metric space. In contrast, LOF and the Yose-Ue treap ensemble display a broader density concentration near the ideal point $\vec{z}^*$, indicating improved average joint performance but increased heterogeneity across benchmarks. To quantify these differences, we report the average anomaly detection gains relative to SOTA baselines in Table 6 and introduce a distance-to-ideal stability metric. As before, we compute the Euclidean distance $d_b$

Table 5: Anomaly detection performance ($\mu \pm \sigma$) of all SOTA methods and Yose-Ue for all benchmarks. The training/testing split is 95/5; the number of treaps is 5, with a subsampling rate of 256.

| Datasets | Metrics | GATE | AE | HBOS | LOF | Yose-Ue (Global, FD) | Yose-Ue (Local, Sturge) |
|---|---|---|---|---|---|---|---|
| C1 | AUCROC | $0.940 \pm 0.025$ | $0.748 \pm 0.057$ | $0.955 \pm 0.000$ | $0.940 \pm 0.000$ | $0.956 \pm 0.003$ | $0.956 \pm 0.004$ |
| | AUCPR | $0.851 \pm 0.050$ | $0.633 \pm 0.045$ | $0.891 \pm 0.000$ | $0.914 \pm 0.000$ | $0.890 \pm 0.014$ | $0.871 \pm 0.015$ |
| C2 | AUCROC | $0.965 \pm 0.037$ | $0.907 \pm 0.021$ | $0.931 \pm 0.000$ | $0.994 \pm 0.000$ | $0.936 \pm 0.021$ | $0.887 \pm 0.071$ |
| | AUCPR | $0.904 \pm 0.048$ | $0.853 \pm 0.027$ | $0.844 \pm 0.000$ | $0.979 \pm 0.000$ | $0.676 \pm 0.086$ | $0.666 \pm 0.126$ |
| SIN | AUCROC | $0.898 \pm 0.038$ | $0.606 \pm 0.056$ | $0.798 \pm 0.031$ | $0.974 \pm 0.000$ | $0.830 \pm 0.014$ | $0.804 \pm 0.024$ |
| | AUCPR | $0.825 \pm 0.054$ | $0.436 \pm 0.046$ | $0.580 \pm 0.053$ | $0.963 \pm 0.013$ | $0.676 \pm 0.055$ | $0.539 \pm 0.027$ |
| FC | AUCROC | $0.377 \pm 0.095$ | $0.959 \pm 0.009$ | $0.654 \pm 0.000$ | $0.999 \pm 0.000$ | $0.937 \pm 0.038$ | $0.860 \pm 0.063$ |
| | AUCPR | $0.123 \pm 0.016$ | $0.783 \pm 0.039$ | $0.294 \pm 0.000$ | $0.997 \pm 0.000$ | $0.687 \pm 0.147$ | $0.491 \pm 0.149$ |
| SMTP | AUCROC | $0.815 \pm 0.071$ | $0.813 \pm 0.040$ | $0.765 \pm 0.000$ | $0.924 \pm 0.000$ | $0.742 \pm 0.114$ | $0.687 \pm 0.122$ |
| | AUCPR | $0.372 \pm 0.180$ | $0.656 \pm 0.003$ | $0.094 \pm 0.000$ | $0.363 \pm 0.000$ | $0.121 \pm 0.137$ | $0.068 \pm 0.100$ |
| PEN | AUCROC | $0.890 \pm 0.077$ | $0.987 \pm 0.007$ | $0.922 \pm 0.000$ | $0.999 \pm 0.000$ | $0.942 \pm 0.027$ | $0.924 \pm 0.044$ |
| | AUCPR | $0.767 \pm 0.098$ | $0.964 \pm 0.021$ | $0.821 \pm 0.000$ | $0.999 \pm 0.000$ | $0.847 \pm 0.066$ | $0.806 \pm 0.103$ |
| THY | AUCROC | $0.900 \pm 0.059$ | $0.948 \pm 0.010$ | $0.989 \pm 0.000$ | $0.982 \pm 0.000$ | $0.966 \pm 0.022$ | $0.972 \pm 0.013$ |
| | AUCPR | $0.815 \pm 0.102$ | $0.931 \pm 0.012$ | $0.976 \pm 0.000$ | $0.970 \pm 0.000$ | $0.905 \pm 0.074$ | $0.900 \pm 0.072$ |
| ST | AUCROC | $0.573 \pm 0.027$ | $0.614 \pm 0.031$ | $0.708 \pm 0.000$ | $0.519 \pm 0.000$ | $0.692 \pm 0.006$ | $0.684 \pm 0.020$ |
| | AUCPR | $0.798 \pm 0.010$ | $0.782 \pm 0.015$ | $0.852 \pm 0.000$ | $0.736 \pm 0.000$ | $0.844 \pm 0.005$ | $0.840 \pm 0.011$ |
| ST-2 | AUCROC | $0.997 \pm 0.001$ | $0.998 \pm 0.001$ | $0.973 \pm 0.000$ | $0.995 \pm 0.000$ | $0.987 \pm 0.002$ | $0.986 \pm 0.002$ |
| | AUCPR | $0.992 \pm 0.001$ | $0.993 \pm 0.004$ | $0.941 \pm 0.000$ | $0.984 \pm 0.000$ | $0.963 \pm 0.008$ | $0.948 \pm 0.026$ |
| FC-K | AUCROC | $0.491 \pm 0.062$ | $0.950 \pm 0.025$ | $0.609 \pm 0.000$ | $0.988 \pm 0.000$ | $0.969 \pm 0.016$ | $0.959 \pm 0.027$ |
| | AUCPR | $0.149 \pm 0.018$ | $0.832 \pm 0.059$ | $0.210 \pm 0.000$ | $0.966 \pm 0.000$ | $0.791 \pm 0.102$ | $0.699 \pm 0.148$ |
| ST-K | AUCROC | $0.672 \pm 0.019$ | $0.567 \pm 0.002$ | $0.606 \pm 0.000$ | $0.549 \pm 0.000$ | $0.594 \pm 0.003$ | $0.623 \pm 0.019$ |
| | AUCPR | $0.594 \pm 0.013$ | $0.454 \pm 0.004$ | $0.596 \pm 0.000$ | $0.444 \pm 0.000$ | $0.610 \pm 0.007$ | $0.628 \pm 0.011$ |
| E-CC | AUCROC | $0.997 \pm 0.004$ | $1.000 \pm 0.000$ | $1.000 \pm 0.000$ | $1.000 \pm 0.000$ | $0.997 \pm 0.004$ | $0.999 \pm 0.001$ |
| | AUCPR | $0.999 \pm 0.001$ | $1.000 \pm 0.000$ | $1.000 \pm 0.000$ | $1.000 \pm 0.000$ | $0.999 \pm 0.003$ | $1.000 \pm 0.000$ |
| E-MF | AUCROC | $0.509 \pm 0.466$ | $1.000 \pm 0.000$ | $1.000 \pm 0.000$ | $1.000 \pm 0.000$ | $1.000 \pm 0.000$ | $1.000 \pm 0.000$ |
| | AUCPR | $0.767 \pm 0.225$ | $1.000 \pm 0.000$ | $1.000 \pm 0.000$ | $1.000 \pm 0.000$ | $1.000 \pm 0.000$ | $1.000 \pm 0.000$ |
| E-VI | AUCROC | $0.828 \pm 0.113$ | $1.000 \pm 0.000$ | $1.000 \pm 0.000$ | $1.000 \pm 0.000$ | $0.984 \pm 0.015$ | $0.993 \pm 0.007$ |
| | AUCPR | $0.950 \pm 0.035$ | $1.000 \pm 0.000$ | $1.000 \pm 0.000$ | $1.000 \pm 0.000$ | $0.993 \pm 0.006$ | $0.997 \pm 0.003$ |

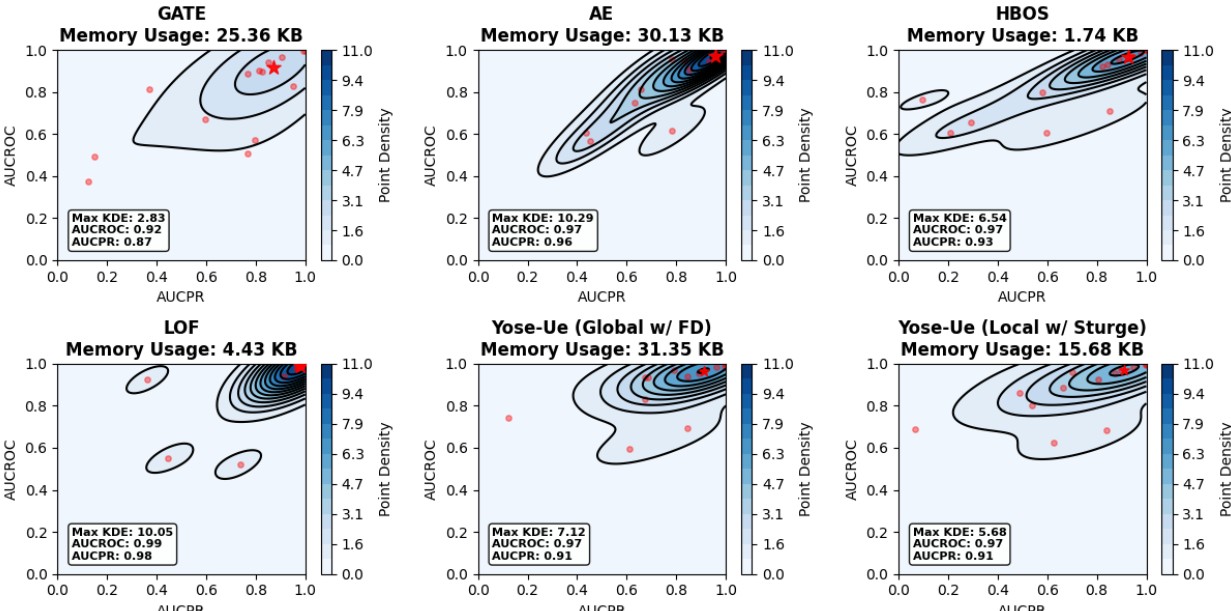

Figure 17: Performance contour plot of all SOTA methods and Yose-Ue. The training/testing split is 95/05.

from each benchmark's joint performance point to the ideal coordinate. To assess stability across datasets, we define the coefficient of variation of this distance as $\mathrm{CV}_d = \sigma_{d_b}/\mu_{d_b}$. Lower $\mathrm{CV}_d$ values indicate greater consistency in proximity to the ideal performance point across benchmarks. In practical anomaly detection settings, such stability is desirable, as it reduces the likelihood of server regressions on individual datasets and improves generalization robustness. We therefore report the reduction in $\mathrm{CV}_d$ achieved by Yose-Ue relative to each SOTA baseline. As shown, the proposed treap ensemble consistently improves AUCROC and reduces $\mathrm{CV}_d$ compared to SOTA methods, while AUCPR gains are less uniform. This behavior can be

Table 6: Average anomaly detection improvement for the Yose-Ue treap Ensemble method compared with SOTA resource-constrained unsupervised anomaly detection techniques. The subsampling size $\psi$ is 256, and the number of estimators is 5. Both AUCPR and AUCROC are reported as improvements for training/testing splits of 95/5, 90/10, and 80/20.

| Train/Test Split | SOTA | Yose-Ue (Global with FD) | | | Yose-Ue (Local with Sturge) | | |
|---|---|---|---|---|---|---|---|
| | | $\Delta$AUCROC | $\Delta$AUCPR | $CV_d$ reduct. | $\Delta$AUCROC | $\Delta$AUCPR | $CV_d$ reduct. |
| 95/5 | GATE | 0.120 | 0.078 | -0.129 | 0.105 | 0.038 | -0.073 |
| | AE | 0.031 | -0.022 | 0.019 | 0.016 | -0.061 | 0.075 |
| | HBOS | 0.044 | 0.064 | 0.036 | 0.030 | 0.025 | 0.092 |
| | LOF | -0.023 | -0.093 | 0.655 | -0.037 | -0.132 | 0.711 |
| 90/10 | GATE | 0.120 | 0.051 | 0.022 | 0.105 | 0.016 | 0.052 |
| | AE | 0.034 | -0.042 | 0.061 | 0.019 | -0.07 | 0.091 |
| | HBOS | 0.050 | 0.051 | 0.118 | 0.035 | 0.015 | 0.148 |
| | LOF | -0.132 | -0.018 | 0.710 | -0.033 | -0.167 | 0.740 |
| 80/20 | GATE | 0.099 | 0.026 | 0.068 | 0.086 | -0.004 | 0.041 |
| | AE | 0.026 | -0.066 | 0.074 | 0.013 | -0.096 | 0.048 |
| | HBOS | 0.040 | 0.015 | 0.185 | 0.027 | -0.014 | 0.158 |
| | LOF | -0.028 | -0.182 | 0.745 | -0.041 | -0.212 | 0.719 |

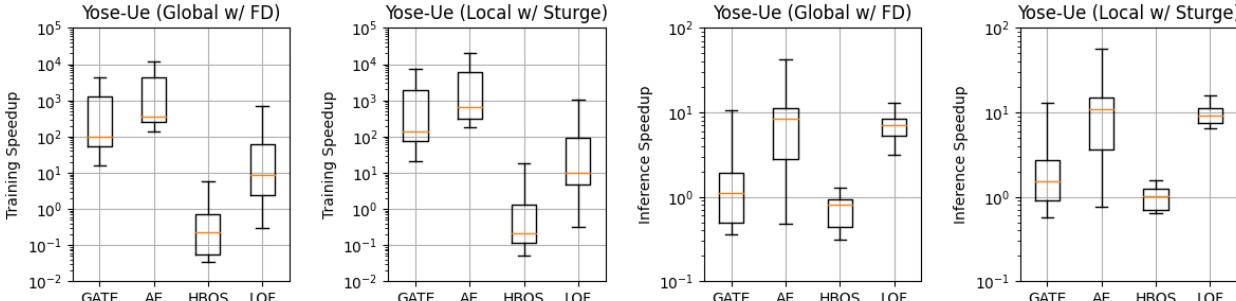

Figure 18: Training time and inference latency speedup across all validation results for the 95/5 train/test split.

Table 7: Training/inference speedups of Yose-Ue variants compared with DNN and non-DNN SOTA methods. Only five estimators were considered within the treap ensemble with a subsampling rate of 256.

| Yose-Ue Type | Average Training Time Speedup | | | | Average Inference Latency Speedup | | | |
|---|---|---|---|---|---|---|---|---|
| | GATE | AE | HBOS | LOF | GATE | AE | HBOS | LOF |
| Global with FD | 979.190 | 2814.327 | 0.887 | 126.068 | 2.062 | 11.371 | 0.739 | 7.150 |
| Local with Sturge | 1708.931 | 4899.083 | 2.088 | 217.151 | 2.615 | 15.718 | 1.008 | 9.877 |

attributed to characteristics of unsupervised anomaly detection in imbalanced regimes. AUCROC primarily reflects ranking quality, whereas AUCPR emphasizes precision at high confidence thresholds (McDermott et al., 2024). Although Yose-Ue improves nominal-anomaly separability, it produces slightly greater score dispersion, which can increase false positives at certain operating points and modestly degrade precision-recall performance. Future work may therefore explore improved thresholding and calibration strategies to further enhance treap ensemble performance in precision-sensitive settings.

Beyond detection accuracy, we analyze the distribution of computational speedups achieved by Yose-Ue during both training and inference. The results are presented in Figure 18. The two leftmost columns report training-time speedup, while the two rightmost columns report inference-latency speedup, each measured relative to SOTA baselines. A value of $10^0$ corresponds to a speedup factor of $\times 1.0$ (i.e., no change in latency), values greater than $10^0$ indicate speedup, and values less than $10^0$ indicate slowdown. To ensure fairness, all models were evaluated independently under identical hardware conditions. As shown, the proposed treap ensemble provides substantial computational gains compared to DNN-based methods and LOF. The average training-time and inference-latency speedups are summarized in Table 7. For example, relative to the DNN autoencoder baseline, Yose-Ue achieves comparable anomaly detection performance while delivering more than $\times 2800$ faster training and over $\times 11$ faster inference. HBOS achieves lower training and inference latency than Yose-Ue; however, Yose-Ue consistently provides superior anomaly detection performance. Conversely, while LOF attains stronger precision–recall performance in certain cases, Yose-Ue offers significantly

improved computational efficiency, yielding over $\times 126$ speedup in training and more than $\times 7$ speedup in inference latency. Overall, these results demonstrate that Yose-Ue achieves a favorable performance–efficiency tradeoff, maintaining competitive detection accuracy while substantially reducing computational cost.

## 12 Conclusion

Anomaly detection is a fundamental capability in modern computing systems, particularly in resource-constrained edge environments where deviations from nominal behavior must be detected efficiently, reliably, and without supervision. Although recent advances in deep learning and ensemble-based approaches have improved detection performance, their computational and memory requirements have often limited their practicality for edge deployment. Furthermore, many existing methods rely on supervised training paradigms that assume access to labeled anomalies—an assumption rarely satisfied in real-world anomaly detection scenarios. In this work, we addressed these challenges by co-designing memory-efficient data representations and compute-efficient unsupervised learning algorithms. We introduced Yose-Ue, a treap-based ensemble anomaly detection framework inspired by the Japanese forest-bundling technique. By combining discretized split points with mass-driven, priority-based split selection, Yose-Ue improved estimator diversity and distributional fidelity while substantially reducing memory overhead. This design preserved the structure of nominal data distributions even under severely constrained ensemble sizes, improving both stability and generalization. We evaluated Yose-Ue across 14 datasets spanning synthetic distributions, experimental iPhone measurements, and benchmark datasets from the ODDS repository. The results demonstrated improved or competitive performance relative to established unsupervised ensemble baselines, including Isolation Forest (iForest), DiForest, and EXTiForest, as well as resource-efficient deep learning and classical methods such as autoencoders, graph attention autoencoders, HBOS, and LOF. In particular, Yose-Ue consistently improved joint anomaly-detection performance (AUCROC and AUCPR) compared with BST-based ensemble methods while maintaining competitive results against state-of-the-art alternatives. Beyond detection accuracy, Yose-Ue provided substantial computational advantages, achieving orders-of-magnitude reductions in training time and significant improvements in inference latency. These properties made the framework well-suited for deployment in memory- and compute-constrained environments. Overall, this work demonstrated that data-structure-aware ensemble learning—specifically treap-based mass estimation—provided a principled and efficient approach to unsupervised anomaly detection. More broadly, the results suggested that revisiting classical data structures in the design of machine learning algorithms can unlock new efficiency–performance trade-offs, enabling more practical solutions for edge-scale system monitoring and resource-constrained machine learning.

### Broader Impact Statement

We have introduced a memory-efficient ensemble learning framework for robust unsupervised anomaly detection under stringent resource constraints. By significantly reducing memory consumption and computational overhead, Yose-Ue enables statistically reliable inference on embedded and edge platforms where conventional ensemble and deep learning approaches have often been impractical. This design facilitates the deployment of monitoring and diagnostic systems in safety- and reliability-critical domains—including infrastructure monitoring, embedded sensing, and secure computing—where timely detection of anomalous behavior is essential for maintaining resilience and operational integrity.

The framework has emphasized structural efficiency through algorithm–data structure co-design, thereby promoting more sustainable edge machine learning by lowering computational demand, energy consumption, and deployment costs at scale. More broadly, this work has advanced scalable and interpretable anomaly detection by demonstrating that statistical robustness can be achieved without sacrificing feasibility in constrained environments. In doing so, it has helped bridge the gap between theoretical reliability and practical implementation in real-world systems.

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
