# OpenReview forum: "Yose-Ue: A Treap-Based Ensemble Framework for Resource-Efficient Unsupervised Anomaly Detection"
_TMLR — Under review for TMLR_

### Review · Reviewer_Unbr · 2026-04-25

**Summary Of Contributions:**

**Summary**

This work proposes Yose-Ue, a novel algorithm for performing unsupervised anomaly detection in resource-constrained deployments. The proposed method builds upon tree ensemble–based approaches by incorporating a treap structure, enabling both training and inference procedures to better capture feature-wise data density.

The authors theoretically demonstrate that, under certain regimes, the proposed method is more efficient in both training and inference compared to conventional randomized BST-based structures. Empirically, the method also achieves superior performance over existing tree ensemble–based anomaly detection approaches.

Furthermore, the proposed approach exhibits significantly faster training and inference speeds than recent DNN–based methods, while maintaining comparable performance.

**Strength**
- The idea of incorporating a treap structure into a tree ensemble framework to construct inverse-mass–maximized split points is highly plausible and well-motivated.
- The paper theoretically identifies regimes in which the proposed algorithm is more efficient than conventional randomized BST-based methods, and the clear illustration in Figure 7 is particularly helpful for understanding this advantage.
- The comparisons presented in Figures 13 and 14, highlighting training variance and peak performance relative to existing methods, are intuitive and facilitate an effective evaluation of the model.
- The proposed algorithm demonstrates clear advantages over DNN-based approaches in terms of training and inference speed.

**Weakness**

While the proposed algorithm is shown to be effective in modeling various probability distributions, as illustrated in Figure 11, and performs well on the datasets presented in Section 5, there may exist classes of distributions that are inherently difficult for the method to capture.

In particular, for spiky multimodal distributions, the algorithm may produce multiple regions with shallow tree depths despite relatively high local density, due to its structural characteristics. This suggests potential limitations in accurately reflecting density in such cases.

**Audience:**

Yes

**Audience Explanation:**

This paper addresses the problem of unsupervised anomaly detection, a topic of strong interest to the TMLR audience. The proposed approach, which integrates a heap-based structure into a tree-based algorithm to tackle the problem in resource-constrained environments, is likely to provide valuable insights and inspiration for researchers working on tree-based methods for tabular data.

**Broader Impact Concerns:**

This paper proposes an algorithm for unsupervised anomaly detection, which contributes to AI safety. As such, no significant ethical concerns or negative societal impacts are anticipated.

**Claims And Evidence:**

Yes

**Claims Explanation:**

The theoretical analysis presented in the paper clearly demonstrates the efficiency of the proposed algorithm, and the experimental evaluations provide strong and reliable support. Together, these theoretical and experimental results convincingly substantiate the characteristics, efficiency, and effectiveness of the proposed method.

**Requested Changes:**

**[Analysis of Distributional Suitability]**

It would be beneficial if the paper could more clearly characterize the types of distributions for which the proposed algorithm performs well versus those where it may be less effective. Such an analysis would help practitioners better understand when to apply the method to specific datasets.

Additionally, it would be helpful to discuss whether these distributional characteristics can explain the observed performance differences of the proposed algorithm across the datasets evaluated in Section 6.


**[Justification and Impact of Discretization Choices]**

The paper proposes two discretization functions based on Sturges' Rule and Freedman–Diaconis Rule, and I am curious about the justification for adopting these particular choices. While the paper provides experimental comparisons between these two variants, it would be helpful to include a clearer explanation of why they were selected as discretization candidates and how their respective characteristics influence the performance of the proposed algorithm.

Additionally, further discussion on why each rule is more suitable for local versus global settings would improve the clarity and completeness of the paper.


**[Ablation of Scoring Function and Training Contributions]**

The proposed anomaly scoring method appears to be applicable to existing tree ensemble algorithms. It would be helpful to evaluate the performance of combining the proposed scoring function with standard tree ensemble methods, as this would allow for a clearer understanding of the respective contributions of the proposed training procedure and the scoring mechanism.

---

> ### Author Response · Authors · 2026-06-24
> **Unbr (R1 Rebuttal Comment 1)**
>
> # Rebuttal to Unbr (R1)
>
> We thank the reviewer for their comments and feedback on our work. We have taken the comments and recommended action items into account. A point-by-point response is provided below for the reviewer's scrutiny. In addition, we identified an error in our values in Table 4. The values reported were from an ensemble of one estimator, rather than five as indicated. We apologize for this and have corrected our values reported in the manuscript. This correction does not change the qualitative conclusions drawn from Table 4 or the corresponding discussion.
>
> ## Weaknesses
> > ``While the proposed algorithm is shown to be effective in modeling various probability distributions, as illustrated in Figure 11, and performs well on the datasets presented in Section 5, there may exist classes of distributions that are inherently difficult for the method to capture.
> > In particular, for spiky multimodal distributions, the algorithm may produce multiple regions with shallow tree depths despite relatively high local density, due to its structural characteristics. This suggests potential limitations in accurately reflecting density in such cases.''
>
> We thank the reviewer for this observation. We agree that sharply multimodal or rapidly varying densities can be challenging when the discretization is too coarse. Yose-Ue does not rely directly on tree depth in its scoring function, but its mass estimates are still limited by the resolution of the discretized feature partitions. We have clarified this in Section 4.2: shallow regions are not a limitation of path-depth scoring itself, but they may reflect insufficient discretization granularity for resolving narrow high-density modes.
>
> We clarify this point in Section 4.2 with the following added to the manuscript: ``Finally, we note that the treap ensemble’s ability to resolve complex distributional structures—particularly sharp or multimodal densities—is governed by the discretization strategy used to generate candidate split points. While the inverse-mass $\max$ split rule ensures a consistent reverse-density ordering within each node, the resolution of this ordering depends on the discretized feature space. Data-driven discretization schemes (e.g., Sturges or Freedman–Diaconis) control this fidelity: finer partitions enable better separation of narrow, high-density modes. Consequently, regions that appear structurally shallow under coarse discretization do not indicate a limitation of density modeling but reflect the chosen partition resolution.''

---

> > ### Author Response · Authors · 2026-06-24
> > **Unbr (R1 Rebuttal Comment 2)**
> >
> > ## Analysis of Distributional Suitability
> > > ``It would be beneficial if the paper could more clearly characterize the types of distributions for which the proposed algorithm performs well versus those where it may be less effective. Such an analysis would help practitioners better understand when to apply the method to specific datasets.
> > >Additionally, it would be helpful to discuss whether these distributional characteristics can explain the observed performance differences of the proposed algorithm across the datasets evaluated in Section 6.''
> >
> > We thank the reviewer for this suggestion. Yose-Ue is well-suited for distributions whose nominal structure can be effectively captured via mass-based partitioning, i.e., heterogeneous or moderately multimodal densities. In cases involving sharp or varying-density regions, performance depends strongly on the discretization resolution needed to resolve high-density modes. This perspective also helps explain trends observed across datasets in Section 6 (now Section 7): datasets with clearer mass separation and stable local structure tend to yield more consistent gains, whereas those requiring finer partition granularity exhibit increased sensitivity. Highlighting these relationships emphasizes the role of discretization for the proposed method.
> >
> > We provide further clarification in Section 6.1 (now Section 7). The added text to the manuscript explaining this is as follows: ``Yose-Ue performs most effectively on datasets where the nominal distribution exhibits a stable mass structure that can be captured through discretized partitioning, including heterogeneous and moderately multimodal densities (e.g., C1, C2). In such cases, the mass-aware treap construction can better represent local geometry and yield stable anomaly scores. Conversely, datasets characterized by extremely sharp or rapidly varying density regions may require finer discretization to fully resolve underlying structure (e.g., ST-K, FC-K). These distributional characteristics help explain the observed empirical trends: datasets with well-separated mass structure exhibit consistent performance improvements, whereas those with high-frequency density variation show increased sensitivity to discretization granularity. Note that this characterization should be interpreted as an empirical guideline rather than a formal distributional guarantee.  However, these results highlight the importance of selecting appropriate discretization strategies when applying Yose-Ue in practice.''
> >
> >
> > ## Justification and Impact of Discretization Choices
> > > ``The paper proposes two discretization functions based on Sturges' Rule and Freedman–Diaconis Rule, and I am curious about the justification for adopting these particular choices. While the paper provides experimental comparisons between these two variants, it would be helpful to include a clearer explanation of why they were selected as discretization candidates and how their respective characteristics influence the performance of the proposed algorithm.
> > > Additionally, further discussion on why each rule is more suitable for local versus global settings would improve the clarity and completeness of the paper.''
> >
> > We thank the reviewer for this question. Sturges’ Rule and the Freedman–Diaconis Rule were selected as discretization strategies because they are well-established [1, 2, 3], statistically grounded methods for determining binning resolution and align with Yose-Ue's resource-efficient, parameter-free design goals. We leverage their complementary characteristics by applying Sturges’ Rule for local discretization and Freedman–Diaconis for global discretization. Sturges’ Rule produces coarser, more stable partitions that reduce sensitivity to noise and sampling variability within nodes, whereas Freedman–Diaconis adapts bin width based on the interquartile range, providing finer, data-driven resolution to capture global distributional spread and heterogeneity. This principled trade-off between stability and expressivity directly influences Yose-Ue’s ability to model diverse data distributions.
> >
> > We provide further clarification in Section 6.1 (now Section 7): ``In this work, we adopt Sturge's rule for local discretization to provide stable local partitioning. The Freedman Diacnois' rule is employed for global discretization, enabling finer data-adaptive resolution across the feature space.''
> >
> > [1] J. M. De La Rubia, "Rice University Rule To Determine The Number of Bins", Open Journal of Statistics, 2014
> >
> > [2] H. A. Sturges, "The choice of a class interval", Journal of the American Statistical Association, 1926
> >
> > [3] D. Freedman, P. Diacnois, "On this histogram as a density estimator: L2 Theory," Zeitschrift fur Wahrscheinlichkeitstheorie und Verwandte Gebiete, 1981

---

> > > ### Author Response · Authors · 2026-06-24
> > > **Unbr (R1 Rebuttal Comment 3)**
> > >
> > > ## Ablation of Scoring Function and Training Contributions
> > > > ``The proposed anomaly scoring method appears to be applicable to existing tree ensemble algorithms. It would be helpful to evaluate the performance of combining the proposed scoring function with standard tree ensemble methods, as this would allow for a clearer understanding of the respective contributions of the proposed training procedure and the scoring mechanism.''
> > >
> > > We thank the reviewer for this suggestion. To further examine the portability of the proposed scoring rule, we added an ablation study that applies the traversal-based score to randomized BST ensembles. A new section and two new figures are constructed to aid in this analysis.
> > >
> > > Below is the added text to the manuscript (now found as Section 10): ``To further highlight the contributions of the proposed treap construction and path-traversal scoring components, we construct a hybrid randomized BST ensemble that utilizes the proposed treap traversal-based scoring strategy. Figure 15 illustrates the behavior of the scoring mechanism on a Normal distribution, while Figure 16 presents critical difference (CD) diagrams computed from the expected AUCROC and AUCPR rankings across all benchmark datasets compared with the proposed treap framework and SOTA forest-based methods benchmarked previously, i.e., DiForest, EXTiForest, and iForest.
> > >
> > > The synthetic experiment highlights a key distinction between conventional isolation-based scoring and the proposed traversal-based mass estimate. Under the iForest heuristic, path depths exhibit substantial variability and require large ensembles to approximate the underlying data distribution. In contrast, the proposed path-traversal score produces a smoother empirical mass profile that more closely follows the shape of the underlying normal distribution (Figure 15). As the ensemble size increases, the traversal score converges toward a stable estimate of relative feature-space mass. However, when we compare the treap-based estimation (Figure 11) with the modified randomized BST ensemble (Figure 15), the randomized BST variants exhibit a wider range of scoring behavior than the proposed treap ensemble. Therefore, this suggests that the treap-construction technique (with inverse-mass priority-based splits) helps reduce our observed mass-estimation uncertainty.
> > >
> > > To determine whether these observations translate into improved anomaly detection performance, we evaluate the ablated variants against the state-of-the-art tree-ensemble baselines and the full treap models. We used average-rank and critical difference analyses based on the expected AUCROC ($\mu_{\rm AUCROC}$) and the expected AUCPR ($\mu_{\rm AUCPR}$) per dataset. Interestingly, randomized BST ensembles equipped with the proposed traversal score outperform DiForest, which is also based on discretized partitions of the feature space. This suggests that the proposed scoring mechanism captures meaningful information about the underlying distribution's shape, thereby improving anomaly detection performance. However, we observed that the randomized BST variants, without treap construction, did not achieve the same performance rankings as the original iForest or the full treap-based models. This indicates that while the scoring rule improves distributional interpretability (as shown in Figure 15), the treap construction improves estimator stability and therefore, contributes to the improved anomaly-detection performance.
> > >
> > > In addition, we observed that across AUCROC and AUCPR, the proposed treap learner framework (Yose-Ue) achieves the best average ranks among all compared methods (Figure 15). In particular, the global Freedman--Diaconis discretization consistently attains the highest ranking, followed closely by the local Sturges-based variant. These results suggest that both components matter: the traversal score improves the information captured by standard randomized BST ensembles, while the inverse-mass treap construction improves estimator stability. The CD analysis supports a consistent ranking advantage for Yose-Ue, although not all close comparisons are statistically distinguishable.''

---

### Review · Reviewer_u9Me · 2026-05-17

**Summary Of Contributions:**

This paper introduces an unsupervised anomaly detection framework tailored for resource-constrained edge environments. The core contribution lies in the co-design of data structures and algorithms. The proposed method combines a discretized feature representation with a randomized treap and includes an inverse-mass split selection mechanism. It achieves competitive detection performance while significantly minimizing both training time and memory overhead.

Strengths:

1.	The method demonstrates remarkable resource efficiency. It achieves substantial computational savings in both training time and inference latency compared to deep learning and classical baselines.

2.	The paper directly tackles a realistic application field, using machine learning solutions directly onto ultra-low-memory edge nodes and microcontrollers where standard deep architectures are completely impractical.

3.	The experimental suite is extensive, evaluating the method across 14 synthetic and real-world datasets. The analysis goes beyond simple downstream performance metrics to explore distribution fitting quality via Kolmogorov-Smirnov tests, parameter sensitivity, and multi-objective performance-memory frontiers.

Weaknesses:

1.	The focused scope and scenarios is highly limited. The proposed mechanism is highly constrained by its design. Because it relies heavily on feature-wise domain discretization, its competitive advantage is only observable in resource-constrained, lower-dimensional scenarios. Therefore, the evaluated datasets are relatively simple. And the proposed method doesn’t always perform baselines if considering only ood detection performance. While I understand the hardware constraints in embedded systems, I am not very familiar with such environments. Therefore, it remains difficult for me to evaluate whether this niche "micro-ensemble" setup represents a broad or sufficiently significant problem domain within the wider ML community.

2.	The theoretical foundation leans heavily on algorithmic and data-structure complexity (Section 4.7) but lacks rigorous treatment from a statistical learning perspective. For a method positioned as a non-deep, density-based statistical proxy, the absence of mathematical guarantees is notable. Specifically, the paper provides no finite-sample confidence bounds or asymptotic consistency proof to guarantee that the mass-traversal scoring converges to the true density. Relying purely on empirical results (such as the KS test) to validate a statistical method feels incomplete.

**Audience:**

Yes

**Audience Explanation:**

The proposed method is explained clearly with theoretical analysis and experimental results.

**Claims And Evidence:**

Yes

**Claims Explanation:**

See my strengths of summary.

**Requested Changes:**

1.	The authors describe a scenario where resources are strictly limited during both training and inference. While it is highly intuitive that edge devices face inference-phase constraints, it is less obvious why the training phase must be heavily restricted to local, low-resource hardware rather than being offloaded to central cloud servers. The authors need to elaborate on the exact target use cases that mandate rapid, on-device local training to solidify the practical motivation.

2.	The presentation in Section 4 and Section 6 feels overly fragmented. Many micro-subsections of wildly varying lengths are jammed together (e.g., Section 4 contains subsections spanning two full pages alongside others that are less than 10 lines long). I highly recommend the authors restructure these sections to bundle related concepts together and improve the clarity and logical flow of the prose.

---

> ### Author Response · Authors · 2026-06-24
> **u9Me (R1 Rebuttal Comment 1)**
>
> # Rebuttal to u9Me (R1)
>
> We thank the reviewer for their feedback on our work. We have taken the recommended action items into account. A point-by-point response is provided below for the reviewer's scrutiny. In addition, the new manuscript revision has additional material in blue text. In addition, we identified an error in our values in Table 4. The values inserted were from an ensemble of one estimator, rather than five as indicated. We apologize for this and have corrected our values reported in the manuscript. This correction does not change the qualitative conclusions drawn from Table 4 or the corresponding discussion.
>
> ## Weaknesses
> > The focused scope and scenarios is highly limited. The proposed mechanism is highly constrained by its design. Because it relies heavily on feature-wise domain discretization, its competitive advantage is only observable in resource-constrained, lower-dimensional scenarios. Therefore, the evaluated datasets are relatively simple. And the proposed method doesn’t always perform baselines if considering only ood detection performance. While I understand the hardware constraints in embedded systems, I am not very familiar with such environments. Therefore, it remains difficult for me to evaluate whether this niche "micro-ensemble" setup represents a broad or sufficiently significant problem domain within the wider ML community.
>
> We thank the reviewer for this perspective and agree that the proposed method is most relevant in resource-constrained settings. The focus of this work is intentional, as deployed edge applications cannot support modern techniques that require large amounts of memory (e.g., large ensembles, deep reconstruction models) while still requiring local anomaly detection under limited memory, compute, and energy budgets. In this context, the compact micro-ensemble Yose-Ue (evaluated using synthetic, open-source, and edge datasets) is intended to provide a practical design/evaluation framework in which memory efficiency, interpretability, and competitive detection performance are jointly important [1, 2]. We provide this clarification in Section 2.2 of the paper. In addition, we have also revised the surrounding language to avoid implying that Yose-Ue is a consistent, continuous-density estimator. Throughout the manuscript, we now frame the method as estimating relative empirical mass over discretized partitions, which is the quantity used for anomaly ranking.
>
> [1] H. Oufettoul et al., "TinyML Applications, Research Challenges, and Future Research Directions", IEEE Learning and Technology Conference, 2024
>
> [2] R. H. Jhaveri et al., "TinyML for Empowering Low-Power IoT Edge Consumer Devices", IEEE Transactions on Consumer Electronics, 2025

---

> > ### Author Response · Authors · 2026-06-24
> > **u9Me (R1 Rebuttal Comment 2)**
> >
> > ## Weaknesses (Continued)
> > > The theoretical foundation leans heavily on algorithmic and data-structure complexity (Section 4.7) but lacks rigorous treatment from a statistical learning perspective. For a method positioned as a non-deep, density-based statistical proxy, the absence of mathematical guarantees is notable. Specifically, the paper provides no finite-sample confidence bounds or asymptotic consistency proof to guarantee that the mass-traversal scoring converges to the true density. Relying purely on empirical results (such as the KS test) to validate a statistical method feels incomplete.
> >
> > We thank the reviewer for this important point. We agree that the current theoretical discussion emphasizes algorithmic and data-structure complexity rather than statistical-learning guarantees. We position Yose-Ue as a resource-efficient empirical mass-estimation mechanism for anomaly detection, rather than as a complete nonparametric density estimator. We will revise the manuscript to clarify the scope and avoid implying stronger statistical claims than are currently supported, as follows.
> >
> > We have added further clarification in Section 6.2 (now Section 8): ``Note that the KS statistic is used as an empirical diagnostic for distributional alignment with a reference density estimate. In future work, finite-sample confidence bounds may be constructed for treap-learners to determine whether the mass-traversal score converges to the true density.''
> >
> > In addition, we also provided a brief statistical discussion from a fixed-discretization perspective in Section 4.5: ``Note that under a fixed discretization, empirical bin counts converge to their population bin probabilities as the sample size increases. Therefore, Yose-Ue can be interpreted as a proxy for an empirical mass model of the feature space, where finite partitions approximate relative feature mass for anomaly detection. In this view, the bin-level mass estimates used to guide the treap construction converge to their corresponding discretized population quantities, and the ensemble-averaged traversal score reflects relative mass across an ensemble of partition-based structures. This does not establish convergence to the true continuous density. Rather, we clarify that Yose-Ue estimates relative mass over a finite discretization of the feature space, instead of performing full continuous density estimation. A more rigorous treatment of score concentration, stability under subsampling, and discretization rules remains an important direction for future work.''

---

> ### Author Response · Authors · 2026-06-24
> **u9Me (R1 Rebuttal Comment 3)**
>
> ## Requested Changes
>
> > The authors describe a scenario where resources are strictly limited during both training and inference. While it is highly intuitive that edge devices face inference-phase constraints, it is less obvious why the training phase must be heavily restricted to local, low-resource hardware rather than being offloaded to central cloud servers. The authors need to elaborate on the exact target use cases that mandate rapid, on-device local training to solidify the practical motivation.
>
> We thank the reviewer for their comment on our current narrative. We agree that inference-time deployment is the primary motivation of this work. We have revised the manuscript to make this explicit. The training-time analysis is included both for completeness and for secondary deployment scenarios involving local adaptation or fine-tuning. To clarify, the work (Section 1) is aimed at ultra-efficient inference settings. The current work, as it stands, provides an ultra-memory-efficient way to encapsulate nominal input data for anomaly-detection inference. However, to fully characterize our work against state-of-the-art methods, we included a training-specific comparison of our complexity proofs against BST-ensemble methods in Section 4.7.2 (now Section 5.2) and provided a training speedup breakdown compared with DNN and classical anomaly detection techniques in Section 6.4 (now Section 11). In conjunction, as the reviewer pointed out, resource-constrained edge training is an additional example use case in which efficiency is desirable.
>
> We provide a specific remote-sensing example in Section 2.3 as part of the background/prior work: ``Although this work focuses on ultra-efficient inference, the same efficiency principles can also support edge-specific training and fine-tuning. In many deployed settings, the bottleneck is not only inference latency, but also the ability to adapt models under strict energy, memory, bandwidth, and compute limits. Remote sensing provides a clear example: tasks such as wildfire detection, disaster response, and hyperspectral anomaly detection require timely decisions when transmitting full-resolution data to the cloud may be costly or infeasible. Onboard AI systems such as ESA's $\Phi$sat-2 demonstrate the value of in-orbit processing for applications such as wildfire detection (ESA, 2025a;b), while onboard preprocessing and data-prioritization methods reduce downlink cost by filtering or selecting useful imagery before transmission (Chatar et al., 2024; Qi et al., 2018). However, remote sensing data can vary across sensors, geographies, seasons, atmospheric conditions, illumination, and event types. This creates a need for lightweight local adaptation, where efficient training or fine-tuning can update deployed models without requiring full cloud-scale retraining (Marti Escofet et al., 2026).''
>
> > The presentation in Section 4 and Section 6 feels overly fragmented. Many micro-subsections of wildly varying lengths are jammed together (e.g., Section 4 contains subsections spanning two full pages alongside others that are less than 10 lines long). I highly recommend the authors restructure these sections to bundle related concepts together and improve the clarity and logical flow of the prose.
>
> We thank the reviewer for their feedback. We have restructured our sections to improve the flow of our narrative. To summarize, we combined sections 4.4 and 4.5 (now section 4.4), combined sections 4.6 and 4.8 (now section 4.5), placed section 4.7 into its own section (now Section 5), and made all subsections in Section 6 their own sections. We have updated the appropriate text in our manuscript to reflect this.

---

### Review · Reviewer_HmoL · 2026-06-18

**Summary Of Contributions:**

The paper is well-motivated, addressing the important problem of enabling anomaly detection on resource-constrained edge devices. The proposed method, Yose-Ue, introduces a novel treap-based ensemble framework where anomalies are detected by identifying the lowest-density regions in the data first and isolating them early in the tree structure. This design accelerates the anomaly detection process while keeping memory usage low.

**Audience:**

Yes

**Audience Explanation:**

The paper addresses a critical and practical concern, making anomaly detection accessible on resource-constrained edge devices. The results presented in Figures 13 and 14 clearly demonstrate that Yose-Ue outperforms competing methods particularly in small ensemble regimes, such as with only 5 estimators, which is a realistic constraint for edge deployment. Compared to Isolation Forest-based methods, Yose-Ue achieves better and more consistent joint coverage of both AUCROC and AUCPR across datasets, as evidenced by the higher Max KDE values (7.20 for Global+FD vs. 5.30 for iForest). The evaluation on real edge sensor datasets, car crash, magnetic fluctuations, and vibrations, further validates that this performance advantage holds in practical deployment scenarios, strengthening the paper's real-world applicability claims.

**Broader Impact Concerns:**

There are no broader impact as such from this research

**Claims And Evidence:**

Yes

**Claims Explanation:**

The results in Figure 5, Figure 11, and Table 3 are quite convincing in demonstrating that the proposed treap-based method estimates the true nominal distribution much more accurately than the iForest baseline, with up to a 10× improvement in the Kolmogorov-Smirnov statistic. Figure 11 further confirms that this improvement holds not only for Gaussian distributions but also for Laplacian and Exponential distributions, strengthening the generality of the claim.

The extensive evaluation across 14 datasets is commendable. In particular, the inclusion of real iPhone sensor datasets — covering car crash detection, magnetic fluctuations, and vibrations — is well-aligned with the paper's stated motivation of edge deployment, and the method performs competitively on these datasets while maintaining computational efficiency.

**Requested Changes:**

I am not an expert in classical anomaly detection methods or edge deployment systems, and this review reflects my best assessment given my current level of familiarity with the topic.

**1. Distribution estimation is evaluated only in the univariate setting.**: Figure 5 and Figure 11 demonstrate distribution estimation quality using univariate distributions only. However, real-world anomaly detection problems are inherently multivariate. The paper's core claim is that Yose-Ue approximates a multivariate mass function, yet this claim is not directly validated in higher dimensions. The authors should include at least one multivariate distribution estimation experiment, or provide a discussion of how the method's estimation quality is expected to change as dimensionality increases.

**2. The framing of SOTA baselines deserves reconsideration.** : The paper compares Yose-Ue against methods it labels as state-of-the-art, including deep learning approaches and even an LLM-based anomaly detector. However, positioning these methods as direct competitors may not be entirely fair. Yose-Ue is a lightweight, interpretable, classical ensemble method — a fundamentally different class of model from deep neural networks. It would strengthen the paper to explicitly acknowledge this distinction and reframe the comparison accordingly: Yose-Ue is not trying to beat LLMs in raw accuracy, but rather to offer a practical, interpretable, and resource-efficient alternative where deep learning is simply not deployable. This reframing would make the contribution clearer and the comparisons more honest.

---

> ### Author Response · Authors · 2026-06-24
> **Hmol (R1 Rebuttal Comment 1)**
>
> # Rebuttal to Hmol (R1)
>
> We thank the reviewer for their feedback on our work. A point-by-point response is provided below for the reviewer's scrutiny. In addition, the new manuscript revision has additional material in blue text. In addition, we identified an error in our values in Table 4. The values inserted were from a single estimator, rather than the five indicated. We apologize for this and have corrected our values reported in the manuscript. This correction does not change the qualitative conclusions drawn from Table 4 or the corresponding discussion.
>
> ## Requested Changes
> > Distribution estimation is evaluated only in the univariate setting.: Figure 5 and Figure 11 demonstrate distribution estimation quality using univariate distributions only. However, real-world anomaly detection problems are inherently multivariate. The paper's core claim is that Yose-Ue approximates a multivariate mass function, yet this claim is not directly validated in higher dimensions. The authors should include at least one multivariate distribution estimation experiment, or provide a discussion of how the method's estimation quality is expected to change as dimensionality increases.
>
> We agree with the reviewer that the multivariate claim should be directly supported. We therefore added a multivariate Gaussian experiment in Section 6.2 (now Section 8), visualized in Figure 12, comparing the true multivariate density surface with BST path-depth estimates and the proposed treap path-traversal mass scores. The results show that in the small-ensemble regime, the treap-based traversal score better preserves the shape of the nominal multivariate mass than the BST path-depth baseline does.
>
> We also clarify the scope of this claim: Yose-Ue is not a full joint density estimator. It forms a randomized, feature-wise discretized partition model whose path scores aggregate mass information across selected features. Consequently, we expect estimation quality to degrade as dimensionality increases, especially when anomaly structure is highly correlated, rotated away from coordinate axes, or concentrated on low-dimensional manifolds. This is consistent with our intended deployment regime: low- to moderate-dimensional edge sensor data where memory and latency constraints dominate.
>
> > The framing of SOTA baselines deserves reconsideration. : The paper compares Yose-Ue against methods it labels as state-of-the-art, including deep learning approaches and even an LLM-based anomaly detector. However, positioning these methods as direct competitors may not be entirely fair. Yose-Ue is a lightweight, interpretable, classical ensemble method — a fundamentally different class of model from deep neural networks. It would strengthen the paper to explicitly acknowledge this distinction and reframe the comparison accordingly: Yose-Ue is not trying to beat LLMs in raw accuracy, but rather to offer a practical, interpretable, and resource-efficient alternative where deep learning is simply not deployable. This reframing would make the contribution clearer and the comparisons more honest.
>
> We agree that Yose-Ue should not be framed as a direct raw-accuracy competitor to large deep models or LLM-based anomaly detectors. We revised Section 11 to distinguish between accuracy-oriented and deployment-oriented baselines. In particular, we exclude methods that require specialized accelerator hardware, such as LLM-based detectors, and compare against DNN and non-DNN methods in resource-constrained settings. The revised framing is that Yose-Ue provides a practical, interpretable, and memory-efficient alternative when deep models are not deployable, rather than a universal replacement for deep anomaly detectors in unconstrained environments.